# Evaluation of Arctic sea-ice drift and its dependency on near-surface wind and sea-ice conditions in the coupled regional climate model HIRHAM-NAOSIM

Xiaoyong Yu[1, 2], Annette Rinke[1], Wolfgang Dorn[1], Gunnar Spreen[3], Christof Lüpkes[4], Hiroshi Sumata[5], Vladimir M. Gryanik[4]

[1] Alfred Wegener Institute, Helmholtz Centre for Polar and Marine Research, Potsdam, Germany

[2] School of Marine Sciences, Nanjing University of Information Science and Technology, Nanjing, China

[3] Institute of Environmental Physics, University of Bremen, Bremen, Germany

[4] Alfred Wegener Institute, Helmholtz Centre for Polar and Marine Research, Bremerhaven, Germany

[5] Ocean and sea ice, Norwegian Polar Institute, Tromsø, Norway

*Correspondence to*: Xiaoyong Yu (xiaoyong.yu@awi.de)

**Abstract.** We examine the simulated Arctic sea-ice drift speed for the period 2003-2014 in the coupled Arctic regional climate
model HIRHAM-NAOSIM 2.0. In particular, we evaluate the dependency of the drift speed on the near-surface wind speed and sea-ice conditions. Considering the seasonal cycle of the Arctic basin averaged drift speed, the model reproduces the summer-autumn drift speed well, but significantly overestimates the winter-spring drift speed, compared to satellite-derived observations. Also, the model does not capture the observed seasonal phase lag between drift and wind speed, but the simulated drift speed is more in phase with the near-surface wind. The model calculates a realistic negative correlation between drift
speed and ice thickness and between drift speed and ice concentration during summer-autumn when the ice concentration is relatively low, but the correlation is weaker than observed. A daily grid-scale diagnostic indicates that the model reproduces the observed positive correlation between drift and wind speed. The strongest impact of wind changes on drift speed occurs for high and moderate wind speeds, with a low impact for rather calm conditions. The correlation under low-wind conditions is overestimated in the simulations compared to observation/reanalysis data. A sensitivity experiment demonstrates the
significant effects of sea-ice form drag from floe edges included by an improved parameterization of the transfer coefficients for momentum and heat over sea ice. However, this does not improve the agreement of the modeled drift speed/wind speed ratio with observations based on reanalysis data for wind and remote sensing data for sea ice drift. An improvement might be achieved by tuning parameters that are not well established by observations.

## 1 Introduction

Arctic sea ice has experienced a rapid decrease in recent decades (e.g. Serreze and Stroeve, 2015; Stroeve et al., 2012). On the one hand, the observed Arctic sea-ice decline is caused by thermodynamic processes, such as increasing air and ocean temperatures, e.g. due to ocean heat transports into the Arctic (Serreze et al., 2007). On the other hand, dynamical processes, such as a changed sea-ice drift in response to changing wind, ocean currents and sea-ice conditions (e.g., reduction of sea-ice concentration and thickness), play an important role in the redistribution of sea ice (Rampal et al., 2011; Serreze et al., 2007;
Spreen et al., 2011). Except for the fast-ice and shear zones, sea ice moves largely in response to local winds and ocean currents. More than 70% of the variance of the central Arctic pack ice motion is explained by the geostrophic wind alone on the time scale of days to months (Thorndike and Colony, 1982). For compact sea ice in the inner Arctic or sea ice near the coasts, the internal friction can be as large as the forces due to the winds and ocean currents (Leppäranta, 2011). The seasonal Arctic basin-wide sea-ice drift speed, however, is additionally correlated with the sea-ice concentration and thickness: Sea-ice drift

speed decreases with increasing ice concentration when ice concentration is low, and sea-ice drift speed decreases with increasing ice thickness when ice concentration is high (Docquier et al., 2017; Olason and Notz, 2014). In order to understand and project Arctic sea-ice changes, it is vital for climate models that they can realistically capture the observed sea-ice drift and its dependency on the atmospheric and oceanic forcing and on the sea-ice conditions at different time scales. Tandon et al. (2018) showed that only few CMIP5 models capture the observed seasonal cycle of sea-ice drift speed and that some models show a seasonal cycle of sea-ice drift in phase with the near-surface wind speed.

The first aim of this paper is to evaluate the simulated sea-ice drift speed and its relation with sea-ice concentration, thickness and near-surface wind speed in the coupled Arctic regional climate model HIRHAM-NAOSIM 2.0 (Dorn et al., 2019). We first evaluate the simulated Arctic basin-wide monthly mean drift, then we evaluate the relationship between sea-ice drift speed and sea-ice conditions/wind speed both on Arctic basin-wide, multi-year monthly mean scale and on daily grid scale.

The second aim of this paper is to explore the sensitivity of the simulated ice drift to the parameterization of the atmospheric near-surface transfer coefficients for heat and momentum over sea ice, in which the effect of sea-ice form drag is included. The air-ice drag controls the sea-ice drift under the influence of the wind forcing. It can be separated into the frictional skin drag due to microscale roughness elements on the sea-ice surface and the form drag caused by large structures like pressure ice ridges and floe edges (Arya, 1973, 1975). Most climate models account for the sea-ice skin drag and consider the form drag only by tuning the spatially and temporally invariant air-ice drag coefficient (Castellani et al., 2018; Tsamados et al., 2014). This approach is poorly constrained by the observations and fails to describe the variability of the air-ice drag processes correctly (Andreas et al., 2010; Lüpkes et al., 2012a; Lüpkes and Gryanik, 2015; Tsamados et al., 2014).

Several model studies that include the sea-ice form drag were carried out (e.g. Castellani et al., 2018; Renfrew et al., 2019; Tsamados et al., 2014). Tsamados et al. (2014) implemented a complex sea-ice form drag parameterization based on many sea-ice cover properties (e.g. sea-ice concentration, vertical extent and area of ridges, freeboard and floe draft, and the size of floes and melt ponds) into the stand-alone sea-ice model CICE. Castellani et al. (2018) implemented a simpler sea-ice form drag parameterization that only relies on sea-ice deformation energy and concentration into the coupled ocean-sea ice model MITgcm. Both studies showed improvement in sea-ice drift after the form drag had been included. Recently, Renfrew et al. (2019) implemented an observation-based parameterization of atmospheric form drag caused by floe edges based on suggestions by Lüpkes et al. (2012a), Lüpkes and Gryanik (2015), and Elvidge et al. (2016) into a stand-alone atmosphere model. The simulation results show an improved agreement of mean atmospheric variables and turbulent fluxes with measurements in cold-air outbreak situations over the Fram Strait when form drag is included.

The physical parameterizations suggested by Lüpkes and Gryanik (2015) were constrained by summertime observations over the sea-ice pack and by aircraft observations over the marginal ice zone (MIZ) during winter. Later the parameterizations were once more validated using a larger and independent set of aircraft data obtained from campaigns during different seasons (Elvidge et al., 2016). This validation work concerned the momentum fluxes. The assumptions of Lüpkes and Gryanik (2015) about heat and moisture fluxes over the MIZ could not yet be evaluated by measurements. Thus, further research is necessary on this issue.

Despite the potential uncertainties in terms of the heat and moisture fluxes, the strategy of the present study is to investigate the performance of the parametrization by Lüpkes and Gryanik (2015) without further modification in a coupled atmosphere-ocean-sea ice model. The parametrization by Lüpkes and Gryanik (2015) contains constants that could be fine-tuned in the future to eventually improve the model results. We leave this optimization, however, to future work since it requires a large number of model runs and extensive analysis work that is beyond the scope of this paper focusing on the principal impact of

sea-ice form drag on the model results.

The organization of this paper is as follows: Section 2 describes the model simulations and the observational data as well as the analysis methods used. In Section 3, we present the simulated long-term average Arctic sea-ice drift speed and its dependency on near-surface wind and sea-ice conditions based on both multiyear, Arctic basin-wide scale and daily grid scale. In Section 4, we discuss the sea-ice form drag impact on the atmospheric near-surface turbulent fluxes and on the sea-ice drift speed. Finally, a summary and conclusions are provided in Section 5.

## 2 Data and Analysis

### 2.1 Model and simulations

#### 2.1.1 Coupled atmosphere-ice-ocean regional model

The model used in this study is HIRHAM-NAOSIM 2.0 (Dorn et al., 2019), which consists of the regional atmospheric model HIRHAM5 and the regional ocean-sea ice model NAOSIM (North Atlantic/Arctic Ocean Sea-Ice Model). HIRHAM5 is based on the numerical weather forecast model HIRLAM-7.0 (Undén et al., 2002) and applies the physical parameterizations of the atmospheric general circulation model ECHAM-5.4.00 (Roeckner et al., 2003). Köberle and Gerdes (2003) give a basic description of NAOSIM, while Fieg et al. (2010) describe NAOSIM's fine-resolution model version, which is used in HIRHAM-NAOSIM 2.0. NAOSIM's ocean component is based on the Geophysical Fluid Dynamics Laboratory (GFDL) Modular Ocean Model MOM-2 (Pacanowski, 1996). The sea-ice component is based on the dynamic–thermodynamic sea-ice model described by Harder et al. (1998). The internal ice stress is described by a viscous-plastic rheology according to Hibler (1979). Thermodynamic snow-ice processes are handled using the zero-layer approach by Semtner (1976). Detailed information about HIRHAM-NAOSIM 2.0 is given by Dorn et al. (2019). The model is applied over a circum-Arctic domain at a horizontal resolution of 1/4º (~27 km) in the atmosphere (HIRHAM5) and 1/12º (~9 km) in the ocean (NAOSIM). The ocean-sea ice domain corresponds to the domain shown in Figure 1.

#### 2.1.2 Long-term simulations of the base configuration

A 10-member ensemble of multi-decadal climate simulations for the period 1979 to 2016 were carried out by Dorn et al. (2019) with the base configuration of HIRHAM-NAOSIM 2.0. These ensemble simulations represent the basis for the present study and are referred to as BASE hereafter. The individual ensemble members used the same atmospheric initialization, but applied different ice-ocean initial conditions, which were taken from January 1 of the last 10 years of a preceding 22-year-long coupled spin-up run for the period 1979-2000 (see Dorn et al., 2019, for more details). The regional model simulations were driven by ERA-Interim reanalysis data (Dee et al., 2011; referred to as ERA-I hereafter). ERA-I provided 6-hourly lateral atmospheric boundary conditions as well as daily surface boundary conditions where required (outside the coupling domain). The lateral ocean boundary conditions in the northern North Atlantic were taken from the Levitus climatology (Levitus and Boyer, 1994; Levitus et al., 1994).

#### 2.1.3 Sensitivity experiments

To investigate the model's sensitivity to the momentum and heat transfer coefficients which explicitly include sea-ice form drag contributions, both a control run (CTRL) and a sensitivity experiment (SENS) for one year (year 2007) were performed. CTRL applies the default atmospheric boundary layer parameterization of ECHAM 5.4 (Roeckner et al., 2003) with air-ice momentum and heat transfer coefficients depending only on atmospheric stability. The sea-ice form drag effect is not included.

SENS includes the improved parameterization of air-ice momentum and heat transfer coefficients adapted based on Lüpkes and Gryanik (2015). There, the transfer coefficients depend on sea-ice concentration and include both the sea-ice skin drag and form drag effects caused by the edges of ice floes in the marginal sea ice zone and of leads in the inner Arctic. The parameterization could also account for the effect by melt pond edges, but this additional effect would require the knowledge of the pond fraction on top of the ice, which is not available in HIRHAM-NAOSIM 2.0. The model applies a flux averaging method, which means that the total flux over a surface covered with sea ice of concentration $A$ and with open water $(1-A)$, is the concentration-weighted mean of both contributions. We describe here only the fluxes over sea ice, in which form drag is included in SENS and refer to Roeckner et al. (2003) for the parameterization of fluxes over open water. In CTRL (as in BASE), however, the air-ice momentum transfer coefficient $C_{d,i}$ and the heat transfer coefficient $C_{h,i}$ include only the sea-ice skin drag effect. They are calculated as

$$C_{d,i} = C_{dn,i}\, f_{m,i} \tag{1}$$

$$C_{h,i} = C_{hn,i}\, f_{h,i} \tag{2}$$

where $C_{dn,i}$ ($C_{hn,i}$) are the drag (heat transfer) coefficients under neutral atmospheric stratification over ice and $f_{m,i}$ ($f_{h,i}$) are the stability correction functions over ice to adjust $C_{dn,i}$ ($C_{hn,i}$) based on atmosphere stability. $C_{dn,i}$ is calculated as

$$C_{dn,i} = \frac{k^2}{[\ln(z_L/z_{0,i}+1)]^2} \tag{3}$$

where $k=0.4$ represents the von Karman constant, $z_L$ is the height of the lowest atmospheric model level, and $z_{0,i}$ represents the skin drag roughness length over sea ice. $C_{hn,i}$ is calculated as

$$C_{hn,i} = \frac{k^2}{\ln(z_L/z_{0,i}+1)\ln(z_L/z_{t,i}+1)} \tag{4}$$

where $z_{t,i}$ represents the scalar roughness length over ice. Equations (1) to (4) are common descriptions of air-ice momentum and heat transfer coefficients except that '+1' was added to both $z_L/z_{0,i}$ and $z_L/z_{t,i}$ in equations (3) and (4). This is done in the model to avoid that the argument of the logarithm can go to zero, for which $C_{d,i}$ ($C_{h,i}$) would go to infinity (see also Giorgetta et al., 2013).

In SENS, the new momentum transfer coefficient $\hat{C}_{d,i}$ and the new heat transfer coefficient $\hat{C}_{h,i}$ over ice, which both include skin drag and form drag effects, are calculated as

$$\hat{C}_{d,i} = C_{dn,i}\, f_{m,i} + C_{dn,f}\left[f_{m,i}A + f_{m,w}(1-A)\right]/A \tag{5}$$

$$\hat{C}_{h,i} = C_{hn,i}\, f_{m,i} + C_{hn,f}\left[f_{h,i}A + f_{h,w}(1-A)\right]/A \tag{6}$$

where $C_{dn,f}$ ($C_{hn,f}$) represent the form drag (heat transfer) coefficients related to neutral conditions over ice, $f_{m,w}$ ($f_{h,w}$) are the stability correction functions over water to adjust $C_{dn,f}$ and $C_{hn,f}$ to atmospheric stability. Equations (5) is obtained by combining the equations (6), (52) and (70) by Lüpkes and Gryanik (2015). Equations (6) is obtained by combining the equations (9), (64) and (74) by Lüpkes and Gryanik (2015). After adding '+1' both to $10/z_0$ and $z_L/z_0$ and replacing $z_0$ with $z_{0,f}$ in equation (65) by Lüpkes and Gryanik (2015), $C_{dn,f}$ is calculated as

$$C_{dn,f} = C_{e10}\left[\frac{\ln(10/z_{0,f}+1)}{\ln(z_L/z_{0,f}+1)}\right]^2 A(1-A)^\beta \tag{7}$$

where $C_{e10}$ represents the effective resistance coefficient (both the aerodynamic resistance coefficient of individual floes and the shape factor), $z_{0,f}$ represents the (form) roughness length, and $\beta$ is a constant exponent describing the dependence of cross wind dimension of a floe on $A$. The values of $C_{e10}$, $z_{0,f}$ and $\beta$ are $2.8 \cdot 10^{-3}$, $0.57 \cdot 10^{-3}$ m and 1.1 respectively. The value of $C_{e10}$ is the average given in equations (48) and (49) by Lüpkes and Gryanik (2015). The value of $z_{0,f}$ is an average resulting from measured roughness lengths by various campaigns considered by Andreas et al. (2010), Lüpkes et al. (2012a) and Castellani et al. (2014). Note that this value is not critical for the parametrization. The value of $\beta$ comes from equation (59) by Lüpkes et

al. (2012a).$C_{hn,f}$ is calculated as

$$C_{hn,f} = \frac{C_{dn,f}}{1+C_{a,f}\sqrt{C_{dn,f}}} \tag{8}$$

where

$$C_{a,f} = \frac{1}{k}\ln\left(\frac{z_{0,i}}{z_{t,i}}\right) \tag{9}$$

Equation (8) represents a simple algebraic transformation of equation (60) by Lüpkes and Gryanik (2015) making use of their equations (59) and (61) with $\alpha_f = \alpha$. The skin drag roughness length over sea ice is set to

$$z_{0,i} = 0.69 \cdot 10^{-3}\ \text{m} \tag{10}$$

and the scalar roughness length over sea ice is parameterized as mostly done in the literature as

$$z_{t,i} = \alpha z_{0,i} \tag{11}$$

with

$$\alpha = \exp\left[3.0 - 29.53 z_{0,i}^{0.25}\right] \tag{12}$$

CTRL and SENS simulations comprise each an ensemble of 10 members, which only differ in their ice-ocean initial state. The ice-ocean initial conditions for CTRL and SENS were produced in exactly the same way as for BASE (see Section 2.1.2). ERA-I provided the boundary forcing also as in the BASE simulations.

## 2.2 Datasets for evaluation

### 2.2.1 Sea-ice drift

For the evaluation of the sea-ice drift speed, satellite-based daily sea-ice drift observations from Kimura et al. (2013) (referred to as KIMURA hereafter) are used. There, the improved maximum cross-correlation method (Kimura and Wakatsuchi, 2000, 2004) were applied to detect ice motions based on AMSR-E brightness temperature. The horizontal resolution of the KIMURA dataset is 60 km x 60 km. The uncertainty of the KIMURA data over the Arctic in summer is from 1.12 to 1.47 km d$^{-1}$ and depends on the drift speed (Sumata et al., 2015a). In winter, the uncertainty is at least 50% smaller than in summer and depends on the drift speed too (Sumata et al., 2015b). Although the accuracy of KIMURA drift speed - is lower than that of buoy data, it has a much wider spatial and temporal coverage and is therefore appropriate for regional model evaluation (Sumata et al., 2015a). Another advantage of the KIMURA product is that it provides ice drift data both in winter and summer. More details are given by Kimura et al. (2013) and Sumata et al. (2015a).

In addition, daily sea-ice drift speed from the Pan-Arctic Ice-Ocean Modeling and Assimilation System (PIOMAS; Zhang and Rothrock, 2003) is used. This enables a consistent and simultaneous evaluation of sea-ice drift speed, concentration and thickness, and near-surface wind speed. The PIOMAS data are downloaded from ftp://pscftp.apl.washington.edu/zhang/PIOMAS/data/v2.1/. The mean horizontal resolution in the Arctic is approximately 22 km according to the PIOMAS description given in http://psc.apl.uw.edu/research/projects/projections-of-an-ice-diminished-arctic-ocean/model/. Detailed information about the PIOMAS dataset is given by Schweiger et al. (2011).

### 2.2.2 Sea-ice concentration

For sea-ice concentration (SIC), the NSIDC bootstrap daily SIC over the Northern Hemisphere, Version 3 (https://nsidc.org/data/nsidc-0079) is used. This SIC dataset is based on Nimbus-7 SMMR and DMSP SSM/I-SSMIS passive microwave data and has been generated using a bootstrap algorithm with daily varying tie-points. It is gridded on the 25 x 25 km$^2$ polar stereographic grid and provides an overall retrieval accuracy of approximately 5-10 %, but the uncertainty could be larger due to factors such as the proximity to land or ice edge, melting during summer, the presence of a large fraction of thin ice or melt ponds within a pixel and the presence of stormy weather conditions (Comiso, 2017).

### 2.2.3 Sea-ice thickness

Since Arctic basin-wide long-term sea-ice thickness (SIT) observations are not available, SIT data from PIOMAS are used in this study as a substitute for observational SIT as done in previous studies (Docquier et al., 2017; Johnson et al., 2012; Shu et al., 2015; J. Stroeve et al., 2014). However, we have to recall that PIOMAS ice thicknesses represent simulation results of a coupled ice-ocean model, even though this model is constrained through the assimilation of observed sea-ice concentrations and sea surface temperatures. Schweiger et al. (2011) showed that the PIOMAS ice thickness agrees with ICESat ice thickness retrievals (in the order of 0.1 m mean difference) and that the spatial thickness patterns agree with each other (pattern correlation coefficients > 0.8). However, PIOMAS appears to overestimate thin ice thickness and to underestimate thick ice (Schweiger et al., 2011).

### 2.2.4 Near-surface wind

For the near-surface wind speed (WS), daily 10-m wind speed from ERA-I is used. The ERA-I data were downloaded from the MARS archive at ECMWF and interpolated to the same 0.25º x 0.25º grid as used in the model's atmosphere component HIRHAM5. More information about this dataset is given by Berrisford et al. (2011). The NCEP/NCAR 10-m wind speed with 1.875º x 1.9º horizontal resolution is used to accompany the PIOMAS sea-ice data as this data were used as the wind forcing for PIOMAS. More information about the NCEP/NCAR dataset is given by Kalnay et al. (1996).

### 2.3 Analysis methods

As the different evaluation datasets for sea ice have different spatial resolution, a bilinear interpolation method is used to remap them onto the NAOSIM grid. The common analysis period used in this study is the 12-year-long period 2003-2014 (only December is included for 2012). This limitation is related to the KIMURA data, which are only available since October 2002, and January to November data are missing for 2012 due to the transition from AMSR-E to AMSR-2. We focus on extended summer (JJAS) and winter (DJFM) in this study.

The domain for the basin-wide analysis covers the Arctic Ocean (referred to as the study domain hereafter; enclosed by the purple line in Figure 1). The study domain is defined following Tandon et al. (2018) and excludes the grid points within a distance of 150 km from each coastline.

The simulations are evaluated by means of the commonly used climatological approach. For this, we present multi-year seasonal mean spatial maps and results spatially averaged over the study domain. The model data represent an average of the 10 ensemble members, which were first spatially averaged over the study domain and then time averaged over the period 2003-2014. Following Olason and Notz (2014) and Docquier et al. (2017), we use scatter plots showing Arctic basin-wide and multi-year averaged monthly mean sea-ice drift speed against sea-ice conditions (sea-ice concentration and thickness) to evaluate the relationships between sea-ice drift speed and sea-ice conditions. The linear fit-lines are added in the scatter plots to assist the comparison of the relationship in the model and in the observation/reanalysis. Furthermore, we present evaluation results based on daily data on the grid scale, i.e. for all grid points within the study domain. With this we aim to statistically evaluate the high spatial and temporal variability in the domain, and we represent this by means of box-whisker plots. Therein, the horizontal bar represents the median, the notch represents the 95% confidence interval of the median, the dot represents the mean, the top and bottom of the box represent the 75th and 25th percentiles, and the upper/lower whiskers represent the maximum/minimum values within the 1.5 times interquartile range (IQR) to 75/25 percentiles.

Normalized ensemble mean differences between SENS and CTRL are used to investigate the influence of sea-ice form drag on atmosphere-ice momentum and heat fluxes, sea-ice states and motion and were calculated by dividing ensemble mean differences of SENS minus CTRL with ensemble standard deviations of these differences. Assuming that the differences

between two random simulations are normally distributed around zero, the normalized differences enable a rough estimate of the statistical significance of the differences. Normalized differences greater than 2 (3) or lower than -2 (-3) indicate that the difference is significant on the 95 % (99.7 %) level.

## 3 Evaluation of simulated sea-ice drift speed

First, the skill of the BASE simulated mean sea-ice drift speed (SID) is quantified (Section 3.1). SID is mainly governed by near-surface wind, sea-ice conditions, and ocean currents. The dependency of SID on near-surface wind speed (WS) (Section 3.2) and on sea-ice conditions (Sections 3.3 and 3.4) is evaluated in each case in terms of both the climatological and the daily grid-scale views.

### 3.1 Multi-year mean sea-ice drift speed (SID)

The simulated mean SID shows a distinct spatial pattern with highest drift speed near the ice edges in the Barents Sea, Greenland Sea, and Labrador Sea in winter and over the Alaskan coast in summer (Figure 1a). Compared to the KIMURA dataset, the study-domain-mean bias (RMSE) is 1.72 km $d^{-1}$ (2.12 km $d^{-1}$) in winter and -0.03 km $d^{-1}$ (0.91 km $d^{-1}$) in summer. The model generally overestimates SID in the ice edge zone and north of the Canadian archipelago, the region of thickest ice, with a maximum bias of ca. 6 km $d^{-1}$ in winter and a smaller bias in summer (Figure 1b and 1d). Compared with the uncertainty in the KIMURA sea-ice drift speed provided by Sumata et al. (2015a, b), the model bias in summer is close to or slightly smaller than the uncertainty of the KIMURA data. This indicates that the sign of the model bias in summer SID is uncertain. In winter, however, the model clearly overestimates the SID over the central Arctic and north of the Canada Archipelago and Greenland, even if considering the uncertainty of the KIMURA data. This overestimation of SID in the thick-ice region may be linked to the underestimation of SIT and SIC and overestimation of WS in that region (Suppl. Figures S1-S3). The SIT differences between HIRHAM-NAOSIM 2.0 and PIOMAS can partly be attributed to the feedback between atmosphere and ocean-sea ice in the fully coupled atmosphere-ocean-sea ice model, which is not present in an ocean-sea ice model like PIOMAS. Analysis of the SIT differences between HIRHAM-NAOSIM 2.0 and CryoSat-2 during winter 2010-2014 (Figure S2; see Hendricks and Ricker, 2019, for details about the CryoSat-2 SIT data used here) confirms that HIRHAM-NAOSIM 2.0 underestimates the SIT over the central Arctic and north of the Canada Archipelago and Greenland, at least in winter. The underestimation of SIT and sea-ice volume compared to PIOMAS has been discussed by Dorn et al. (2019) for the longer time period 1979-2016.

The model reproduces the basic mean seasonal cycle of SID of the KIMURA data (Figure 2) with respect to timing and absolute values between May and December. The average bias between June and November is almost zero (-0.0003 km $d^{-1}$) and the corresponding RMSE is 0.28 km $d^{-1}$. The maximum ice drift speed (10 km $d^{-1}$) occurs in October both in the model and in the KIMURA data. However, the model substantially overestimates the SID speed in winter and early spring. The averaged bias (RMSE) between December and May reaches 1.52 km $d^{-1}$ (1.64 km $d^{-1}$). In addition, the modeled minimum SID occurs in May, two months delayed compared to the KIMURA (minimum SID in March). This reflects a longer-lasting and much weaker SID reduction from autumn to spring in the model compared to the KIMURA data. Furthermore, the interannual variation of SID in the model is lower than in the KIMURA data (Figure 2). For example, the interannual amplitude is 0.70 km $d^{-1}$ (1.26 km $d^{-1}$) in the model (KIMURA data) in March and 0.36 km $d^{-1}$ (0.74 km $d^{-1}$) in September, respectively.

The overestimation of winter SID could be related to the underestimation of winter SIT (Figure S2). Besides, the prescribed values of the ice-ocean drag coefficient and the ice strength parameter could also play a role. The ice-ocean drag coefficient, $C_{dw}$, in the base configuration of HIRHAM-NAOSIM 2.0 ($C_{dw} = 5.5 \times 10^{-3}$) is comparable to other CMIP5 models, even though a few models use 3-4 times higher coefficients (see Tandon et al., 2018). Higher ice-ocean drag might damp the SID and its

strong dependency on the wind speed. Docquier et al. (2017) showed that the higher the ice strength parameter, the lower the winter SID and the lower SIT. The ice strength parameter, $P^*$, in the base configuration of HIRHAM-NAOSIM 2.0 ($P^* = 30,000$ N m$^{-2}$) is already slightly higher than in all CMIP5 models (see Tandon et al., 2018). This value has been established for stand-alone ocean-ice simulations with daily wind forcing, but might still be too low considering the hourly wind forcing from the interactively coupled atmosphere.

## 3.2 Dependency of sea-ice drift speed (SID) on near-surface wind speed (WS)

### 3.2.1 Climatological view

As shown in previous studies (Docquier et al., 2017; Kushner et al., 2018; Olason and Notz, 2014; Tandon et al., 2018), the observed distinct mean seasonal cycle of SID (maximum in autumn, minimum in spring) is obviously not solely controlled by the wind speed, which is strongest in winter and weakest in summer (Figure 2). The phase lag between the seasonal cycle of SID and WS is about 3-4 months in observations/reanalysis (KIMURA ice drift/ERA-I wind). The modeled seasonal cycle and magnitude of the WS agrees well with ERA-I. According to the delayed SID minimum (Section 3.1), the phase lag between the simulated seasonal cycle of SID and WS is reduced to about one month, like in many CMIP3 and CMIP5 models (Rampal et al., 2011; Tandon et al., 2018), leading to a higher correlation between SID and WS. This indicates that the modeled SID is much stronger controlled by the wind speed than the observed SID.

Another metric to quantify the mean relationship between SID and WS is the wind factor, which is the ratio of SID to WS. Averaged over the study domain, the simulated wind factor is 1.77% in winter and 1.87% in summer, which agrees with the observations/reanalysis (KIMURA ice drift/ERA-I wind) in the sense that the averaged wind factor is smaller in winter (1.42%) than in summer (1.96%). However, compared to the observations, the simulated wind factor is too large in winter. As mentioned in Section 3.1, too high sensitivity of the SID to the wind in winter may be related to the underestimated SIT and model parameters governing the sea-ice dynamics.

Figure 3 shows the spatial patterns of the multi-year mean wind factor over the Arctic. In winter, the simulated wind factor has its maximum associated with strong wind and low ice thickness over the Baffin Bay and Greenland Sea. Within the study domain, the maximum occurs along the Transpolar Drift Stream (~2%) and the minimum over north of Greenland and the Canadian Archipelago where sea ice is thickest (<0.5%). The simulated pattern roughly agrees with the corresponding pattern of KIMURA ice drift/ERA-I wind (Figure 3 and Suppl. Figure S4) and previous results using SSM/I ice drift/NCEP wind (Spreen et al., 2011). In winter, however, the simulated wind factor is overestimated compared to the KIMURA/ERA-I data almost everywhere over the study domain, with the maximum bias reaching 1% over the thick ice north of the Canadian Archipelago (Figure 3). In summer, the modeled wind factor peaks (~3%) along the marginal ice zone, such as in the coastal Beaufort Sea. This could be the result of a dynamical coupling between sea ice and the coastal ocean as suggested by Nakayama et al. (2012): In a coastal ocean covered with sea ice, wind-forced sea-ice drift excites coastal trapped waves and generates fluctuating ocean currents. These ocean currents can enhance the sea-ice drift when the current direction is the same as the wind-driven drift direction. In contrast to winter, the modeled wind factor in summer is underestimated over the study domain. These seasonally different bias patterns in the wind factor are associated with those of the SID (Figures 1b and 1d).

### 3.2.2 Daily and grid-scale view

Previous studies reported on the ice drift's nearly linear increase with increasing wind for high and moderate WS (Thorndike and Colony, 1982), but no clear relationship for low WS (Leppäranta, 2011; Rossby and Montgomery, 1935). To investigate this in detail, Figures 4 and 5 present the analysis of the dependency of SID on WS based on daily data and on the grid scale. For the observation/reanalysis reference we use KIMURA ice drift and ERA-I wind.

During winter, the simulated SID significantly increases with increasing WS. According to the median values of SID and the associated best fit-line slope of SID vs. WS, the increase of SID varies from 1.29 km d$^{-1}$ to 1.49 km d$^{-1}$ per 1 m s$^{-1}$ WS increase under different SIC classes (Figures 4a and 5a). The strongest SID increase per WS increase occurs for SIC of 50-70% (Figure 4a). In the observations/reanalysis, the SID only consistently increases with increasing WS when SIC is higher than 90% (Figures 4b and 5b). The corresponding slope indicates 0.93 km d$^{-1}$ per 1 m s$^{-1}$ WS increase, which is an approximately 40% smaller increase than the modeled value. Both the model (for all SIC classes) and the observation/reanalysis (for SIC > 90%) show a linear increase of SID with increasing WS for strong winds (WS > 4 m s$^{-1}$), but a weak dependency of SID on WS for lower winds.

During summer, both the model and the observation/reanalysis show a consistent general increase of SID with increasing WS for all SIC classes when WS > 4 m s$^{-1}$ (Figures 4c, 4d, 5d and 5e). The simulated magnitude of SID increase per 1 m s$^{-1}$ WS increase is similar as in winter. Again, the simulated SID increases faster with increasing WS than the observed SID, by a factor of about 2-2.5. Another striking difference between the simulations and the observation/reanalysis is that the simulated SID-WS relation is rather linear(only slightly attenuated when WS $\leq$ 2 m s$^{-1}$). In contrast, this relation is highly nonlinear in the observation/reanalysis (Figure 4d, 5e). The observed increase of SID with increasing WS is much stronger when WS passes a threshold (of ca. 4 m s$^{-1}$), compared to lower WS. For weak winds (WS lower than 4 m s$^{-1}$), SID is not much affected by WS changes.

Generally, the observed SID (for a certain WS and SIC class) shows an about 2 times higher spatiotemporal variability (indicated by the larger IQR; Figure 4) than the modeled one. This discrepancy might be linked to a data source difference or inconsistency between the used KIMURA SID observations and ERA-I reanalysis WS data, but it might also hint to a model weakness.

The PIOMAS-based SID-WS based relation (PIOMAS ice drift, NCEP/NCAR wind) is very similar to that in the HIRHAM-NAOSIM 2.0 simulations: SID increases with increasing WS for all SIC classes, both in winter and summer (Figures 5c and 5f). However, the SID-WS relation in PIOMAS is stronger than in HIRHAM-NAOSIM 2.0 during winter and summer (Figure 5). This may reflect a higher air-ice drag in PIOMAS than in HIRHAM-NAOSIM 2.0.

### 3.3 Dependency of sea-ice drift speed (SID) on sea-ice concentration (SIC)

### 3.3.1 Climatological view

Figure 6a shows the relationship between SID and SIC in terms of the mean seasonal cycle. An inverse correlation between SID and SIC is expected because a reduction in SIC reduces the internal friction, brings the SID closer to free drift conditions and increases SID. We find such an inverse correlation between SID and SIC both in the model and in the observations (KIMURA ice drift, NSIDC bootstrap ice concentration), as well as in PIOMAS from June to September when SIC is low. The corresponding fit line slope in the model is 4.04 km d$^{-1}$ SID increase per 10% SIC decrease, which agrees with the slopes in the observations and in PIOMAS. During the winter months (January to May) when SIC > 95%, such a relationship is not found both in the observations and in the models. The SID of the 12 months can be categorized into two groups based on SIC values. Group one for SIC smaller than 95% and group two for SIC larger than 95%. The observed mean SID value in group one (9.23 km d$^{-1}$) is obviously larger than in group two (7.28 km d$^{-1}$). This is also found in PIOMAS. The model reproduces the SID in group one, but largely overestimates the SID in group two. Therefore, the simulated mean SID values in group one (9.24 km d$^{-1}$) and two (8.95 km d$^{-1}$) are quite close to each other. This model deficit may result from the too strong coupling between the SID and the WS in the model (Figure 2; Section 3.2.1). Following Docquier et al. (2017), we also calculate the slope of the SID-SIC best fit line based on data from 12 months, even though there is no clear SID-SIC linear relation over the

full year data. The slope in the model (0.66 km d$^{-1}$ SID increase per 10% SIC decrease) is much weaker than in the observations (6.93 km d$^{-1}$ SID increase per 10% SIC decrease) and in PIOMAS (3.16 km d$^{-1}$ SID increase per 10% SIC decrease). The simulated relationship is also much weaker than in the ocean model NEMO-LIM3.6 as reported by Docquier et al. (2017). The underestimated relationship in the model can be explained by the overestimated SID in winter (Figure 2).

### 3.3.2 Daily and grid-scale view

On daily grid scale, the model shows an inverse correlation of SID with SIC during winter (Figures 4a, 5a), differently to the climatological view (section 3.3.1). SID increases with decreasing SIC when SIC is larger than ca. 30%. The slopes of the simulated SID-SIC relation across all SIC and WS classes vary from 3.41 km d$^{-1}$ to 5.64 km d$^{-1}$ SID increase per 10% SIC decrease. In the observation, there is no clear SID-SIC relation because the median SID uncertainty is very high when SIC is lower than 90% (Figures 4b, 5b). The small sample size is one of the reasons for the high uncertainty (Table 1). However, considering the uncertainty of the median value (represented by the notch), the statistically lower SID for high SIC (> 90%), compared to the SID for lower SIC, stands out both in the observations and in the simulations. The observations indicate an abrupt drop of SID at 90% SIC. This is in accordance with observations by Shirokov (1977) who showed that SID starts to decrease linearly with increasing SIC, but drops abruptly at a SIC of 90% when internal friction forcing starts its significant influence on SID. This is not reproduced by the simulations and indicates too low internal friction in the model.

The model also shows an inverse correlation of SID with SIC for WS > 2 m s$^{-1}$ during summer (Figures 4c and 5d). Under different WS classes, the modeled SID-SIC relation varies from 2.12 km d$^{-1}$ to 3.16 km d$^{-1}$ SID increase per 10% SIC decrease. The higher WS is, the stronger is the SID-SIC relation. However, these findings from the simulations are not confirmed by the observations. The observations (KIMURA drift/NSIDC bootstrap ice concentration) do not show an inverse correlation between SID and SIC for all SIC and WS classes in summer (Figures 4d and 5e).

The SID-SIC relation in PIOMAS is just the opposite compared to HIRHAM-NAOSIM 2.0, both in winter and in summer: SID increases with increasing SIC when SIC is lower than 90% (Figures 5c and 5f). PIOMAS shows a SID-SIC relation inconsistent with the observed relation. Thus, the PIOMAS relation might violate physical consistency due to the used optimal interpolation method to obtain realistic sea-ice field (concentration). This assimilation method contains addition/subtraction of sea ice into the system at every assimilation time step, when the modeled sea ice concentration differs from the observed one. Due to the addition/subtraction of sea ice (called increment or innovation in the terminology of data assimilation), PIOMAS does not necessarily preserve the physical relations described in the underlying sea ice-ocean model. Such an inconsistency is one of the drawbacks of the optimal interpolation method. Therefore, relations to assimilated physical variables should be examined with caution.

Generally, the obvious difference between the SID-SIC relation on multi-year Arctic mean scale and on daily grid scale emphasizes its strong dependency on the temporal and spatial scale.

### 3.4 Dependency of sea-ice drift speed (SID) on sea-ice thickness (SIT)

### 3.4.1 Climatological view

All datasets (HIRHAM-NAOSIM 2.0 simulations, KIMURA drift/PIOMAS thickness, and PIOMAS drift/PIOMAS thickness) show that SID decreases with increasing SIT (Figure 6b). The SID-SIT fit-line slope calculated based on HIRHAM-NAOSIM 2.0 is 1.10 km d$^{-1}$ SID decrease per 1 m SIT increase. The observed SID-SIT relation is stronger. Data of KIMURA drift/PIOMAS thickness and PIOMAS drift/PIOMAS thickness indicate 2.17 km d$^{-1}$ and 1.83 km d$^{-1}$ SID decrease per 1 m SIT increase, respectively. The simulated weaker dependency of SID on SIT is mainly due to the overestimated SID during winter and spring (December to May), when the ice is thick. As pointed out by Olason & Notz (2014), the inverse correlation between

drift speed and thickness in winter, when the ice concentration is high, is physically plausible, but the inverse correlation in summer, when the ice concentration is lower, is probably only of statistical nature.

### 3.4.2 Daily and grid-scale view

As in the climatological view, the simulations show generally that SID decreases with increasing SIT (Figures 7a and 7c and Suppl. Figure S5a and S5c), both in winter and in summer. In winter, this relation occurs only for SIT smaller than approximately 4 m. The simulated SID-SIT relation that calculated based on all thickness categories varies from 0.38 km d$^{-1}$ to 0.95 km d$^{-1}$ SID decrease per 1 m SIT increase across different wind categories (Suppl. Figure S5a). PIOMAS simulates a stronger SID-SIT relation (Suppl. Figure S5b); the slope of the SID-SIT fit-line varies from 0.65 km d$^{-1}$ to 2.64 km d$^{-1}$ SID decrease per 1 m SIT increase, which is larger by a factor of up to 3.5 compared to the simulations. In summer, the simulated inverse SID-SIT correlation is stronger than in winter, particularly for high WS (Figures 7a and 7c and Suppl. Figures S5a and S5c). The SID-SIT slope varies from 0.71 km d$^{-1}$ to 1.69 km d$^{-1}$ per 1m SIT increase across different wind categories. In PIOMAS, the SID-SIT slopes are not significant when WS > 4 m s$^{-1}$. This is mainly influenced by the increase of SID when SIC increased from (0.1,0.3] to (0.3, 0.5]. Furthermore, Figure 7 confirms the nonlinearity of the SID-WS relation, with a strong impact of WS changes on SID for high and moderate wind speeds, but with a lower impact for low wind speeds (WS $\leq$ 4 m s$^{-1}$).

### 4 Sensitivity to the parameterization including sea-ice form drag

### 4.1 Model results

In the previous section, we investigated the dependency of SID on WS and sea-ice conditions in the BASE simulations, which do not account for the sea-ice form drag. In this section, we analyze how an additional sea-ice form drag influences the near-surface atmospheric fluxes, SID and its dependency on WS and sea-ice condition. For this we compare the one-year ensemble simulations CTRL (without sea-ice form drag) and SENS (with sea-ice form drag) (see Section 2.1.3 for details). Tsamados et al. (2014) showed that an additional form drag mainly improves the sea-ice drift in summer. Therefore, our analysis is focused on summer (JJAS).

Figure 8a indicates that the increased ice roughness due to the additional form drag leads to an increased surface momentum flux over most regions of the Arctic Ocean, even though this increase is not significant from a statistical point of view. In few regions the momentum flux does not increase, which indicates that either the neutral drag coefficient over ice is very low or the wind speed and/or the atmospheric stratification has changed. SID mainly increases over the MIZ along the Russian coasts (Figure 8b). The spatial pattern of SID changes is similar to that of the momentum flux changes, with a pattern correlation of 0.66 (0.76) between their normalized (not normalized) differences (normalized differences were calculated by dividing ensemble mean differences with ensemble standard deviations of these differences, see Section 2.3). This reveals that the SID increase is mainly associated with an increased momentum flux from the atmosphere to the ice. Related with this, WS decreases over most parts of the Arctic Ocean when the sea ice becomes rougher (Figure 8c). Changes in turbulent heat fluxes, SIC, and 2-m air temperature are small and mixed with regional increases and decreases (Figures 8d-f). Overall, the ensemble mean differences between SENS and CTRL are statistically insignificant, indicating a large across- ensemble scatter due to high internally generated model variability.

Nonetheless, the increase of SID for low SIC in the MIZ and the smaller effect on the more consolidated ice in the central Arctic should imply a stronger Arctic-wide SID-SIC relation. Figures 9 and 10a confirm the stronger increase of SID with decreasing SIC in SENS. The increased SIC-dependency of SID can be explained by the fact that the form drag contribution to the air-ice drag is effectively proportional to $(1 - A)^{\beta}$ (see Equations (5) and (7)). This means that the lower the SIC, the

higher the form drag and the higher the air-ice drag. When SIC is around 90%, the sum of form drag and skin drag in SENS is of the same order of magnitude as the skin drag in CTRL.

Furthermore, there is an increased WS-dependency, i.e. that the SID change is larger for high WS. For example, the SID-SIC relation for low wind conditions (WS of 2-4 m s$^{-1}$) in SENS is 5.26 km d$^{-1}$ SID increase per 10% SIC decrease, which is about 2.5 times higher than in CTRL. The differences between SENS and CTRL increase even more for high wind speeds. For WS within 8-10 m s$^{-1}$, the SID-SIC relation in SENS is 11.79 km d$^{-1}$ SID increase per 10% SIC decrease, which is about 4.6 times higher than in CTRL. The dependence of SID on wind is complex. Two external forces act on the flow, the ocean drag and atmospheric drag. When the ocean drag is much larger than the atmospheric drag (low wind speed), the drift is mainly governed by the ocean drag even with doubled atmospheric drag coefficient. However, for high wind speeds, a doubling of the atmospheric drag coefficient has a large effect on the drift speed, also due to proportionality of the air-ice stress to the square of WS.

## 4.2 Model versus observation

The increased dependency of SID on WS and SIC in SENS compared to CTRL does not reduce the deviation to the observation/reanalysis. In contrast, Figure 10 shows that the overestimation of the dependency of SID on WS and SIC in SENS is larger than in CTRL.

Figure 11 shows the spatial distribution of the summer 2007 wind factor, WS and near-surface air temperature from observation/reanalysis data, and the deviations of these three variables in CTRL and SENS from observation/reanalysis. It is obvious that both the bias patterns and the bias magnitudes in CTRL and in SENS are quite similar. Considering the ensemble mean bias and taking the internal model variability into account, it is hard to detect significant changes in SENS, compared to CTRL, as discussed in Section 4.1.

Although the ensemble mean of Arctic basin-wide mean SID from July to September in SENS is larger than in CTRL, the differences are not statistically significant due to a large ensemble spread (Figure 12). Actually, there are no statistically significant differences in the Arctic basin-wide mean SID between CTRL and SENS in all months. From January to May, the simulated Arctic basin-wide mean SID (both in CTRL and in SENS) are higher than that in KIMURA. With respect to the summer months (June to September), the August SID in CTRL is lower than in KIMURA, while the July and September SID in SENS are higher than in KIMURA. For the Arctic basin-wide mean WS, there is no significant difference between CTRL and SENS as well as between model and reanalysis, except for May, when both model simulations significantly overestimate the WS.

## 4.3 Follow-up study

Although the new parameterization does not improve the simulated dependency of SID on WS and sea-ice conditions compared to observations/reanalysis, the sensitivity study clearly shows that the new parameterization does increase the SID due to the added form drag. In a follow-up study, we are going to put efforts therefore on several aspects to improve the simulations. First, tunable parameters of the new parameterization, such as $z_0$, $z_t$, $Ce_{10,i}$, $Ce_{10,k}$ and $\beta$ represent an opportunity to better adapt the form drag parameterization itself to the observations. A first step could be the use of values found by Elvidge et al. (2016). A large effect can be expected by a modification of the skin drag coefficient, since a large region would be affected, and large variations in the drag due to pressure ridges allow a wide range of values. Second, model parameters outside the new parameterization, which have direct impact on SID, like ice strength and ocean-ice drag coefficient, need to be harmonized with the new parameterization, since their values were chosen empirically in terms of adequately balanced performance of the

model. A key is probably the oceanic form drag. Its effect is accounted for in the present study only indirectly via the constant oceanic drag coefficient. Such a parametrization is probably too simple, especially when atmospheric form drag is included (see also Tsamados et al., 2014). Birnbaum (2002) as well as Lüpkes et al. (2012b) found in a mesoscale modelling study that oceanic form drag can have a strong decelerating effect on SID especially when the sea ice concentration is low so that the discussed drawbacks for small sea ice fraction would be reduced or even removed. This effect of form drag on SID was discussed also by Steele et al. (1989). The parametrizations are evidently not balanced anymore after improving one key process of the SID-related atmosphere-ocean-ice interaction. A previous study on the surface-albedo feedback by Dorn et al. (2009) showed that an improved simulation can only be achieved by a harmonized combination of more sophisticated parameterizations of the related sub-processes. It can be assumed that this holds true for the SID-related sub-processes.

## 5 Summary and conclusions

We evaluated the sea-ice drift speed (SID) and its dependency on the near-surface wind and the sea-ice conditions (ice concentration and thickness) on multiyear, Arctic-wide mean scale during 2003-2014. Compared with observations, the model does not fully capture the observed SID seasonal cycle, but overestimates SID in winter-spring. Regardless, the model realistically describes the main drivers of the seasonal and long-term variations of Arctic SID: When the sea-ice concentration (SIC) is lower than 95% in summer-autumn, SID increases with decreasing SIC. However, when SIC is higher than 95% in winter, the sea-ice thickness (SIT) is the main factor for SID changes (higher SID for lower SIT). As the simulated SID is overestimated during the cold seasons, the modeled strength of the SID-SIC and SID-SIT relations is underestimated compared with observational data. The SID overestimation during winter in the model cannot simply be attributed to the underestimation of SIT or too strong coupling between SID and WS. Further in-depth analysis of the wintertime sea-ice dynamics and thermodynamics and the atmospheric and oceanic forcing are needed in the future to identify the possible cause of the SID overestimation.

The analysis on the daily and grid scale revealed that the SID relations with SIC and SIT are complex due to the large spatiotemporal variation of the sea ice. Based on observations, it is difficult to find a clear relation between SID and SIC. In the model, when SIC is larger than 30%, a higher SIC is accompanied by a significantly lower median SID compared to the SID for lower SIC. In agreement with the multi-year and Arctic wide findings, the SID decreases with increasing SIT, both in winter and in summer. However, in winter, this relation occurs only for high wind speeds. The simulated SID-SIT relation is stronger (weaker) in summer (winter) compared to PIOMAS.

We also evaluated the dependency of SID on near-surface wind speed (WS) on the daily and grid scale. The simulated increase of SID with increasing WS is consistent with the observation/reanalysis. Our analysis supports the earlier discussed strong nonlinearity of the SID-WS relation, with a strong impact of WS changes on SID for high and moderate wind speeds, but with only a low impact for lower wind speeds. The weak dependency of SID on WS, when WS is low, was also shown by Lund et al. (2018), based on shipboard marine radar sea-ice drift measurement over the MIZ. This changing relation in the low-wind regime may be caused by the increasing importance of upper-ocean currents for the drift compared to the wind (Lund et al., 2018). Another explanation was given by Leppäranta (2011) and Rossby and Montgomery (1935) who argued that the large-scale wind dominates over the local wind effect.

Finally, we investigated the impact of the changed parameterization of the transfer coefficients for heat and momentum over sea ice on the SID and its dependency on WS and sea-ice conditions. The consideration of sea-ice form drag effects increases the air-to-ice momentum flux and accordingly the SID over most of the Arctic. Largest effects appear for low SIC in the MIZ under high wind conditions. The reason is that in the new parameterization the lower SIC is associated with a potentially larger

atmosphere-ice momentum flux, since the form drag contribution is completely transferred to the ice. As a consequence, the
increases of SID with decreasing SIC is stronger when the form drag is included.

The inclusion of the melt pond effect on the atmospheric form drag in the model might be beneficial. In the current version, form drag was only considered at the edges of ice floes, mainly in the marginal sea-ice zone but not on top of the ice where, melt ponds cause form drag also during summer (Andreas et al., 2010; Lüpkes et al., 2012a). Additional form drag at the ice-
ocean interface may further improve the simulated SID-WS relation, because the oceanic form drag has normally the opposite effect on the ice motion as the atmospheric form drag (Steele et al., 1989; Lüpkes et al., 2012b). Systematic biases in the reanalysis used for the calculation of the 'observed' wind factor cannot be excluded, since form drag is not taken into account in the underlying atmospheric model. Therefore, the increased deviation of the simulated SID-WS relation from the observations/reanalysis does not necessarily mean that the implemented new parameterization worsens the SID-WS relation.


**Data availability.** HIRHAM-NAOSIM data are available at the tape archive of the German Climate Computing Center (DKRZ; https://www.dkrz.de/up/systems/hpss/hpss); one needs to register at DKRZ to get a user account. We will also make the data available via Swift (https://www.dkrz.de/up/systems/swift) on request. KIMURA sea ice drift data are available at https://ads.nipr.ac.jp/vishop/. The ERA‑Interim data were obtained from the European Centre for Medium‑Range Weather
Forecasts (ECMWF; http://apps.ecmwf.int/datasets/data/interim_full_moda/). The sea‑ice concentration data were obtained from the National Snow and Sea Ice Data Center (NSIDC; https://nsidc.org/data/NSIDC‑0051/).

**Author contribution.** X.Y. prepared the manuscript with contributions from all co-authors; Parameterization development: C.L., V.G.; Simulations and Parameterization implementation: W.D.; Assistance with satellite data: H.S., G.S.; Methodology:
X.Y., A.R.; Analysis and Visualization: X.Y.; Writing – original draft: X.Y., A.R.; Writing - improving,editing: all co-authors.

**Competing interests.** The authors declare that they have no conflict of interest.

**Acknowledgments.** We gratefully acknowledge the funding by the Deutsche Forschungsgemeinschaft (DFG, German
Research Foundation) – Project number 268020496 – TRR 172, within the Transregional Collaborative Research Center "ArctiC Amplification: Climate Relevant Atmospheric and SurfaCe Processes, and Feedback Mechanisms (AC)³". X.Y. was supported by the project "Quantifying Rapid Climate Change in the Arctic: regional feedbacks and large-scale impacts (QUARCCS)" funded by the German and Russian Ministries of Research and Education.

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

**Table 1** The sample sizes of sea-ice drift speed data under different 10-m wind speed and sea-ice concentration classes in Figure 4. In the labels of different sea-ice concentration and 10-m wind speed classes, "(" means exclusive and "]" means inclusive.

| Season | Data source | Wind classes | Sea-ice concentration classes | | | | |
|---|---|---|---|---|---|---|---|
| | | | (0.1,0.3] | [0.3,0.5] | (0.5,0.7] | (0.7,0.9] | (0.9,1.0] |
| DJFM | model | (0,2] m/s | 494 | 864 | 3328 | 135516 | 45562216 |
| | | (2,4] m/s | 2070 | 3716 | 12780 | 489893 | 183056161 |
| | | (4,6] m/s | 3432 | 5766 | 17172 | 618383 | 239287363 |
| | | (6,8] m/s | 5804 | 9402 | 17215 | 435238 | 181532612 |
| | | (8,10] m/s | 7950 | 12511 | 18413 | 246921 | 102066710 |
| | KIMURA/ERA-I/NSIDC | (0,2] m/s | 0 | 7 | 7 | 40 | 102295 |
| | | (2,4] m/s | 15 | 29 | 50 | 124 | 365803 |
| | | (4,6] m/s | 32 | 66 | 117 | 279 | 499634 |
| | | (6,8] m/s | 28 | 88 | 137 | 381 | 386667 |
| | | (8,10] m/s | 53 | 83 | 145 | 288 | 218638 |
| JJAS | model | (0,2] m/s | 92547 | 2535519 | 17432992 | 25896542 | 9142130 |
| | | (2,4] m/s | 322269 | 8521163 | 57295068 | 84426782 | 30714941 |
| | | (4,6] m/s | 534300 | 12186383 | 81681898 | 113048227 | 40015962 |
| | | (6,8] m/s | 549254 | 9864436 | 67161887 | 84920918 | 27346273 |
| | | (8,10] m/s | 356102 | 4746320 | 34131702 | 38211228 | 11159363 |
| | KIMURA/ERA-I/NSIDC | (0,2] m/s | 1519 | 3439 | 5150 | 12096 | 100317 |
| | | (2,4] m/s | 4727 | 10504 | 16204 | 39911 | 312814 |
| | | (4,6] m/s | 6533 | 14571 | 22364 | 55418 | 385667 |
| | | (6,8] m/s | 5261 | 11976 | 17587 | 40559 | 262251 |
| | | (8,10] m/s | 2514 | 5249 | 7276 | 17272 | 107988 |


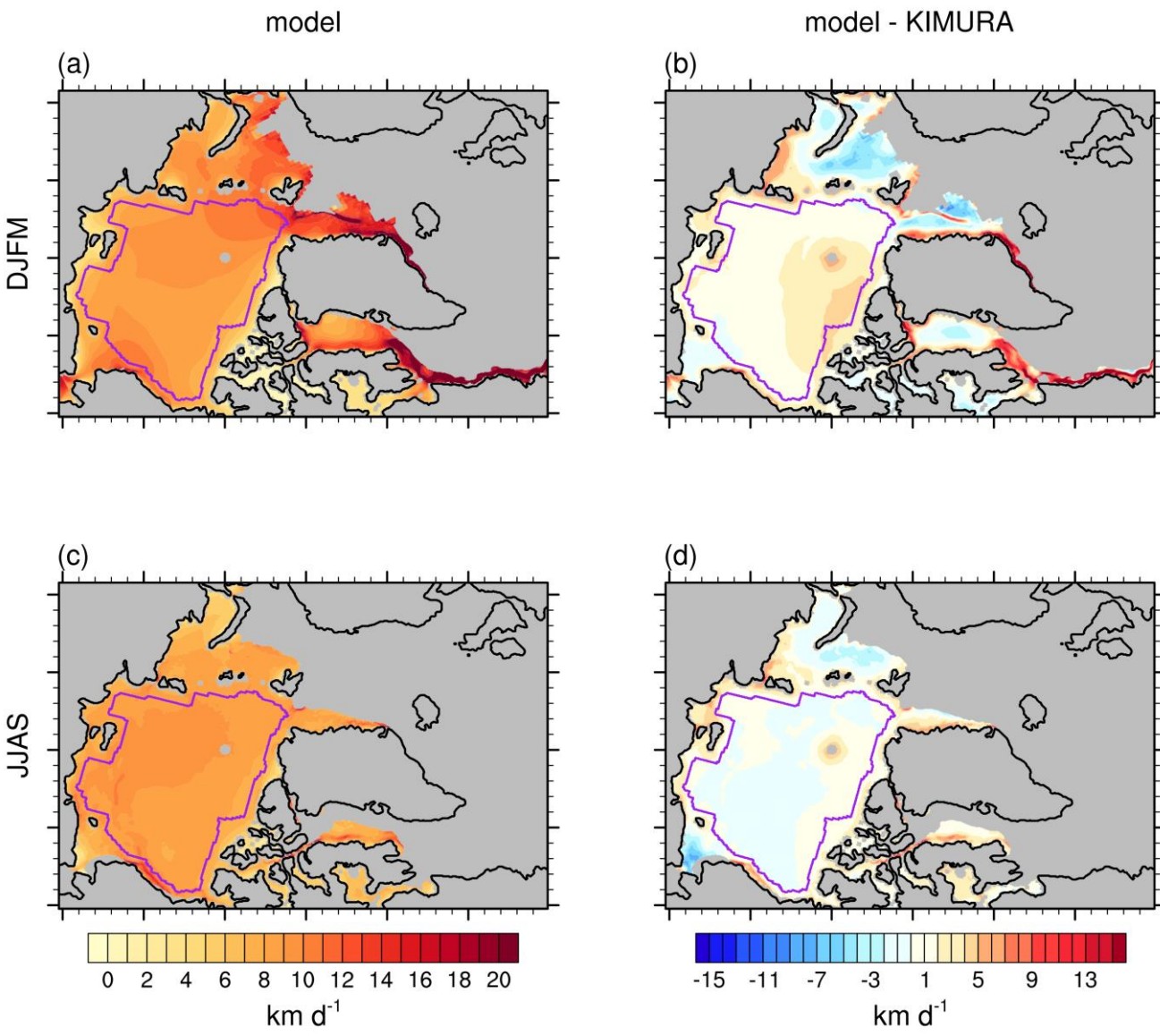

**Figure 1**: Mean spatial pattern of sea-ice drift speed [km d$^{-1}$] in the model (ensemble mean of HIRHAM-NAOSIM 2.0) for 2003-2014 (a) winter (DJFM) and (c) summer (JJAS). (b) and (d) are the model differences to the observation ("Model - KIMURA") for winter and summer respectively. The purple line in each panel indicates the study domain used for the basin-wide analysis.


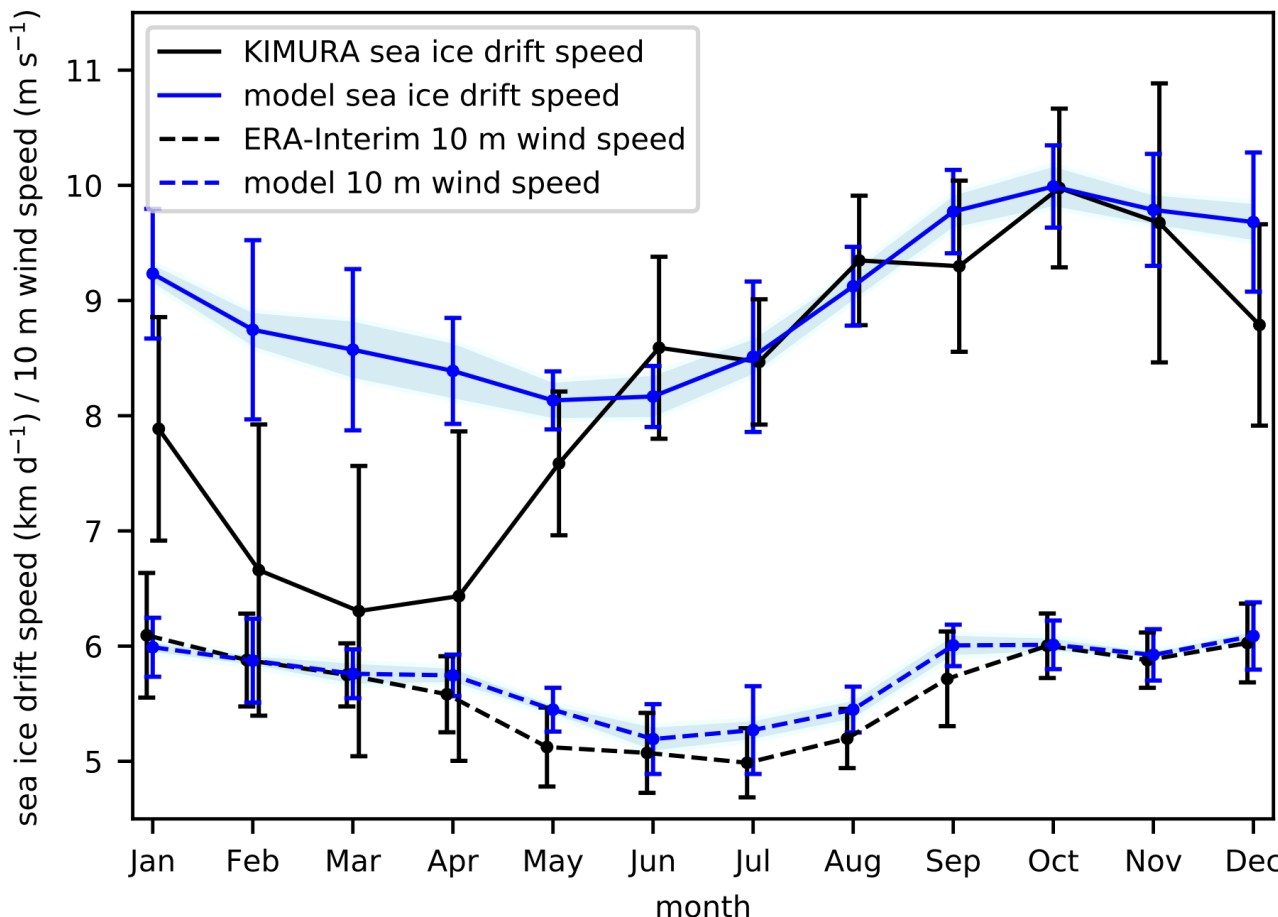

**Figure 2**: Mean annual cycle of sea-ice drift speed [km d$^{-1}$] (solid lines) and 10-m wind speed [m s$^{-1}$] (dashed lines), based on the model (ensemble mean of HIRHAM-NAOSIM 2.0; blue lines) and observation/reanalysis (KIMURA ice drift, ERA-I wind; black lines) for 2003-2014 over the study domain (indicated in Figure 1). The across-ensemble scatter (standard deviation) of the simulations is included as shaded area. The interannual variation is shown by error bars.


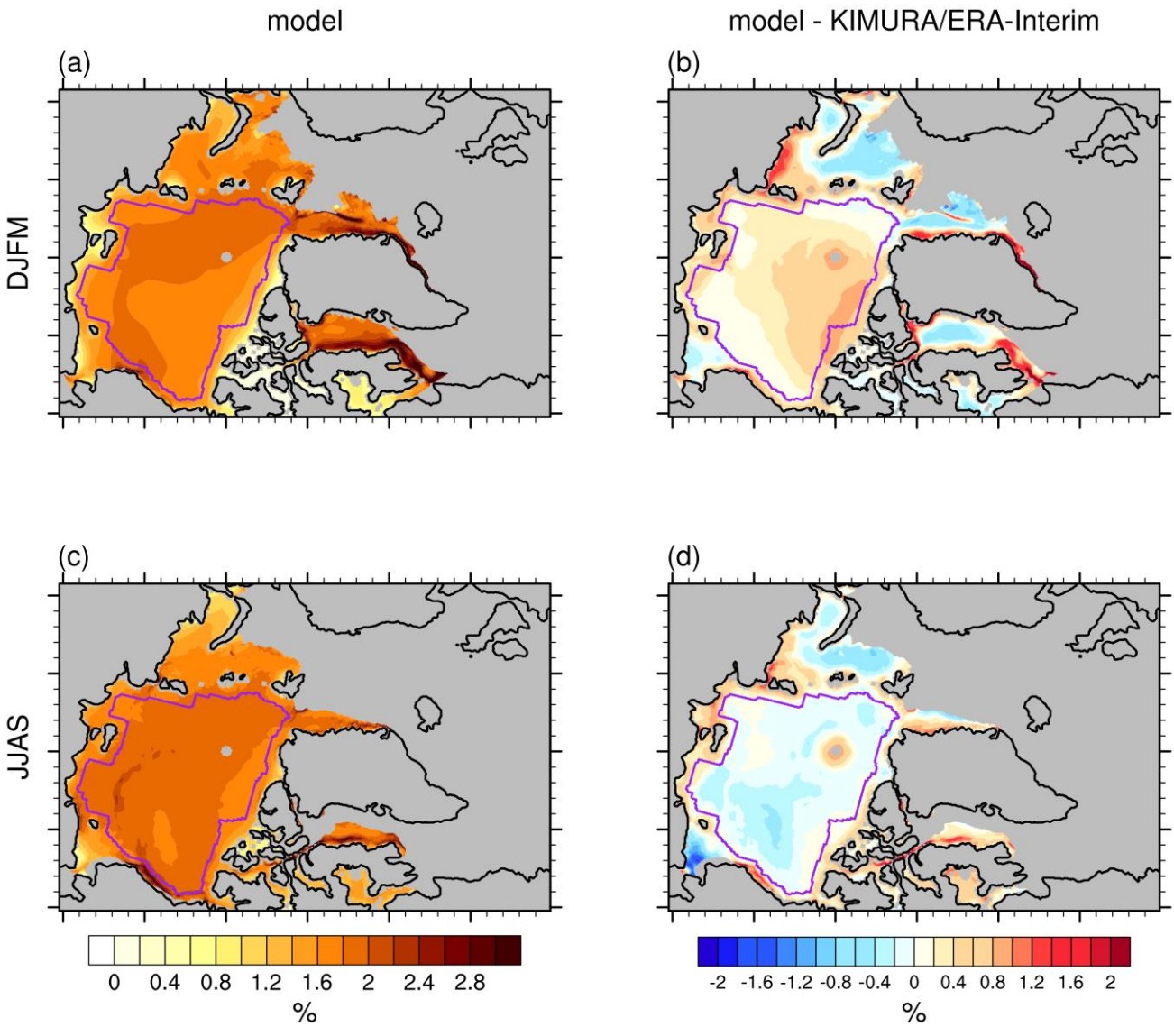

**Figure 3**: Mean spatial pattern of wind factor [%] in the model (ensemble mean of HIRHAM-NAOSIM 2.0) for 2003-2014 (a) winter (DJFM) and (c) summer (JJAS). (b) and (d) are the model differences to the observation/reanalysis ("Model - KIMURA/ERA-I") for winter and summer respectively. The purple line indicates the study domain used for the basin-wide analysis.


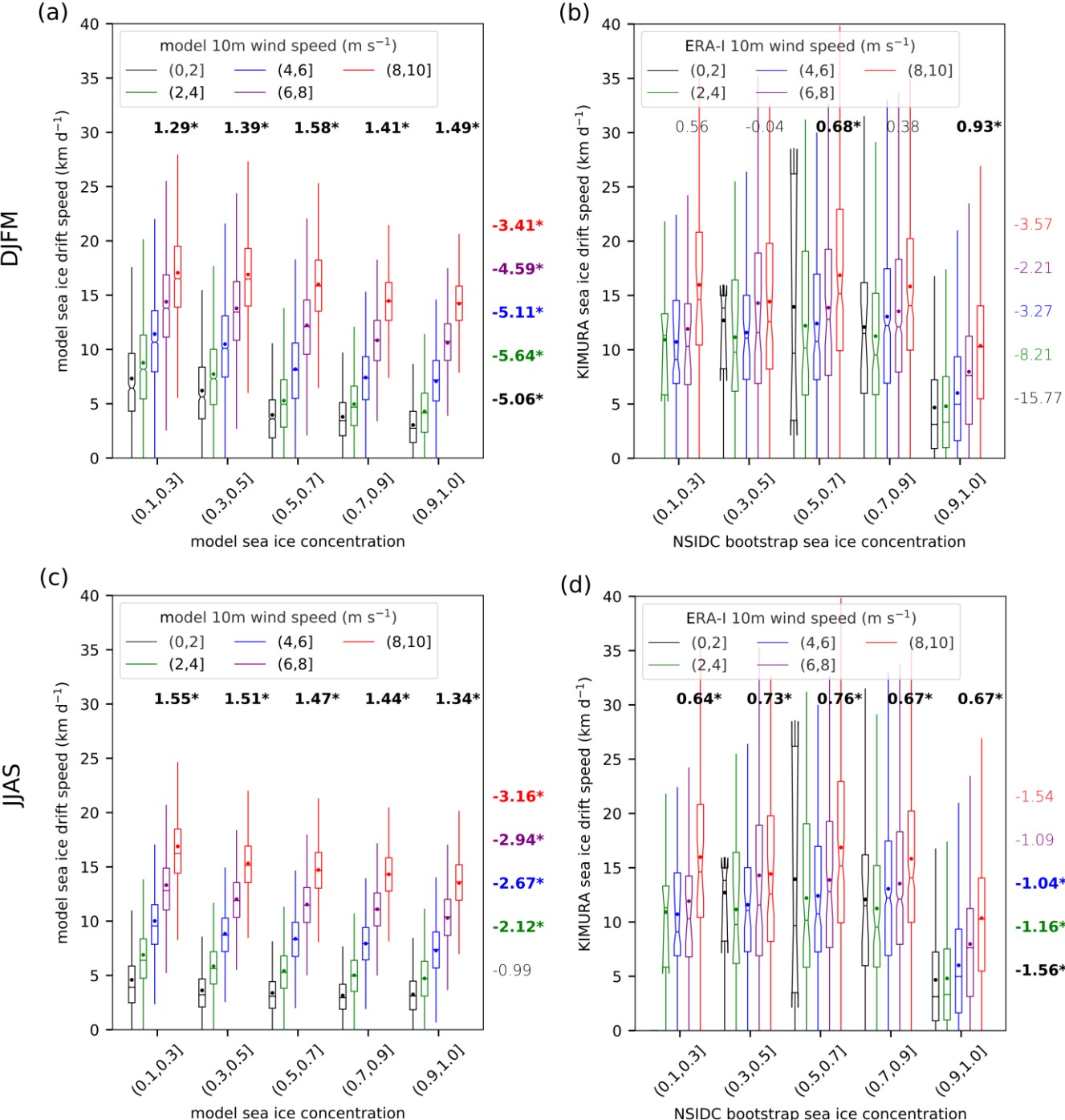

**Figure 4**: Box-whisker plots of the relationship between sea-ice drift speed and sea-ice concentration for different near-surface wind speed classes (different colors) for 2003-2014 (a) winter (DJFM) and (c) summer (JJAS) in the model (HIRHAM-NAOSIM 2.0). (b) and (d) are the same as that for (a) and (c) respectively, but based on the observation/reanalysis data. For the model, all 10 ensemble members are included. The plot is based on daily data and on all grid points within the study domain indicated in Figure 1. The horizontal bar represents the median, the notch represents the 95% confidence interval of the median, the dot represents the mean, the top and bottom of the box represent the 75th and 25th percentiles, the upper/lower whiskers represent the maximum/minimum value within 1.5 times interquartile range (IQR) to 75/25 percentiles. The numbers above the boxplots represent the slopes of near-surface wind and sea-ice drift speed fit lines (unit: km d$^{-1}$ per 1 m s$^{-1}$ wind speed change; font colors as for the wind speed classes). The numbers right of the boxplots represent the slopes of sea-ice concentration and sea-ice drift speed fit lines (unit: km d$^{-1}$ per 10% sea-ice concentration change). A bold and asterisked number indicates that the slope of the fit line is significant at the 95 % level. In the labels of different sea-ice concentration and 10-m wind speed classes, "(" means exclusive and "]" means inclusive.

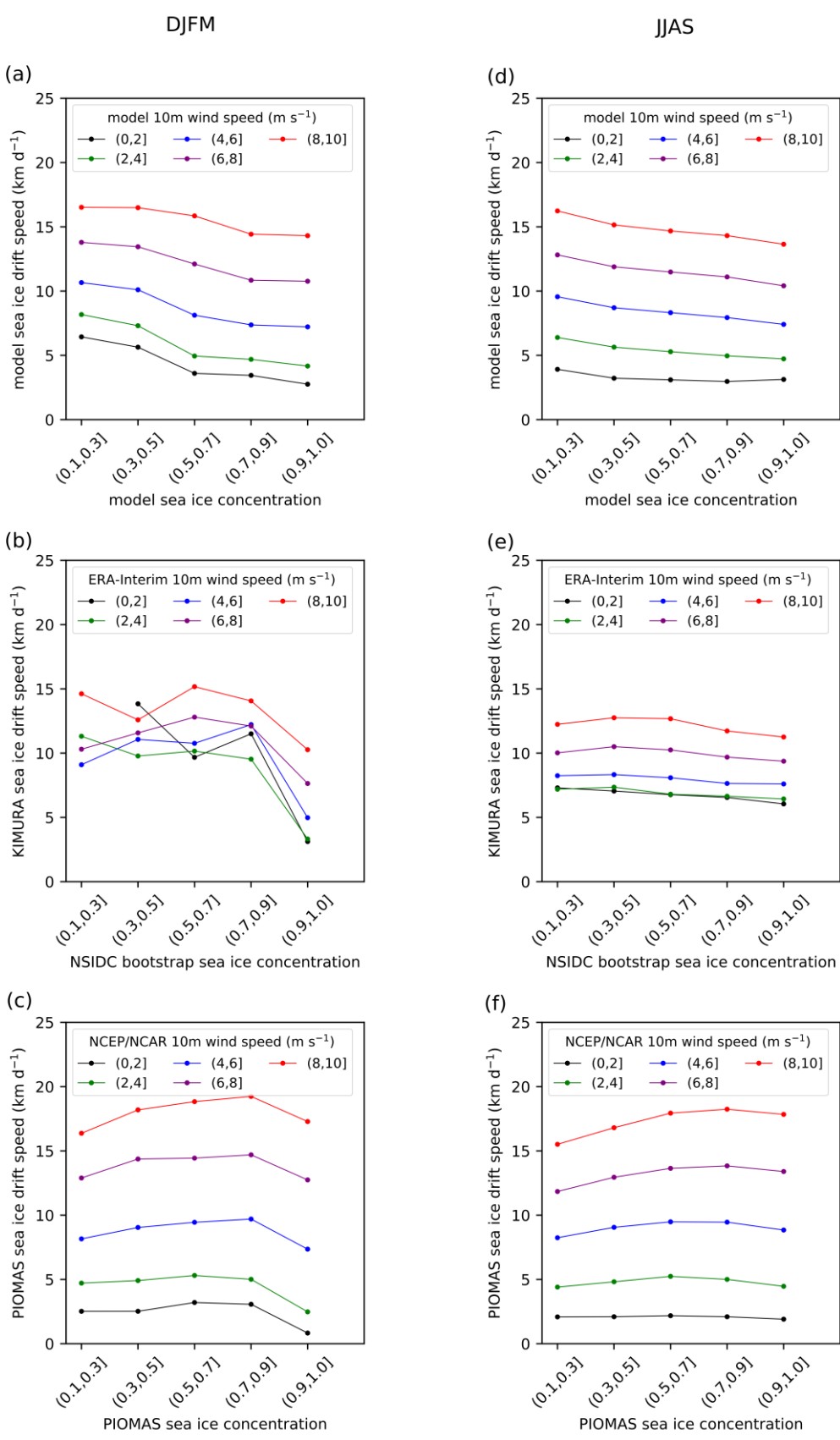

**Figure 5**: Relationship between sea-ice drift speed and sea-ice concentration for different near-surface wind speed classes (different colors) for 2003-2014 (a) winter (DJFM) and (d) summer (JJAS) in the model (HIRHAM-NAOSIM 2.0). (b) and (e) are the same as that for (a) and (d) respectively, but based on the observation/reanalysis data. (c) and (f) are the same as that for (a) and (d) respectively, but based on PIOMAS data. The points in the plot are the median value of all the daily data and on all grid points for certain wind speed and sea-ice concentration, within the study domain indicated in Figure 1. In the labels of different sea-ice concentration and 10-m wind speed classes, "(" means exclusive and "]" means inclusive.

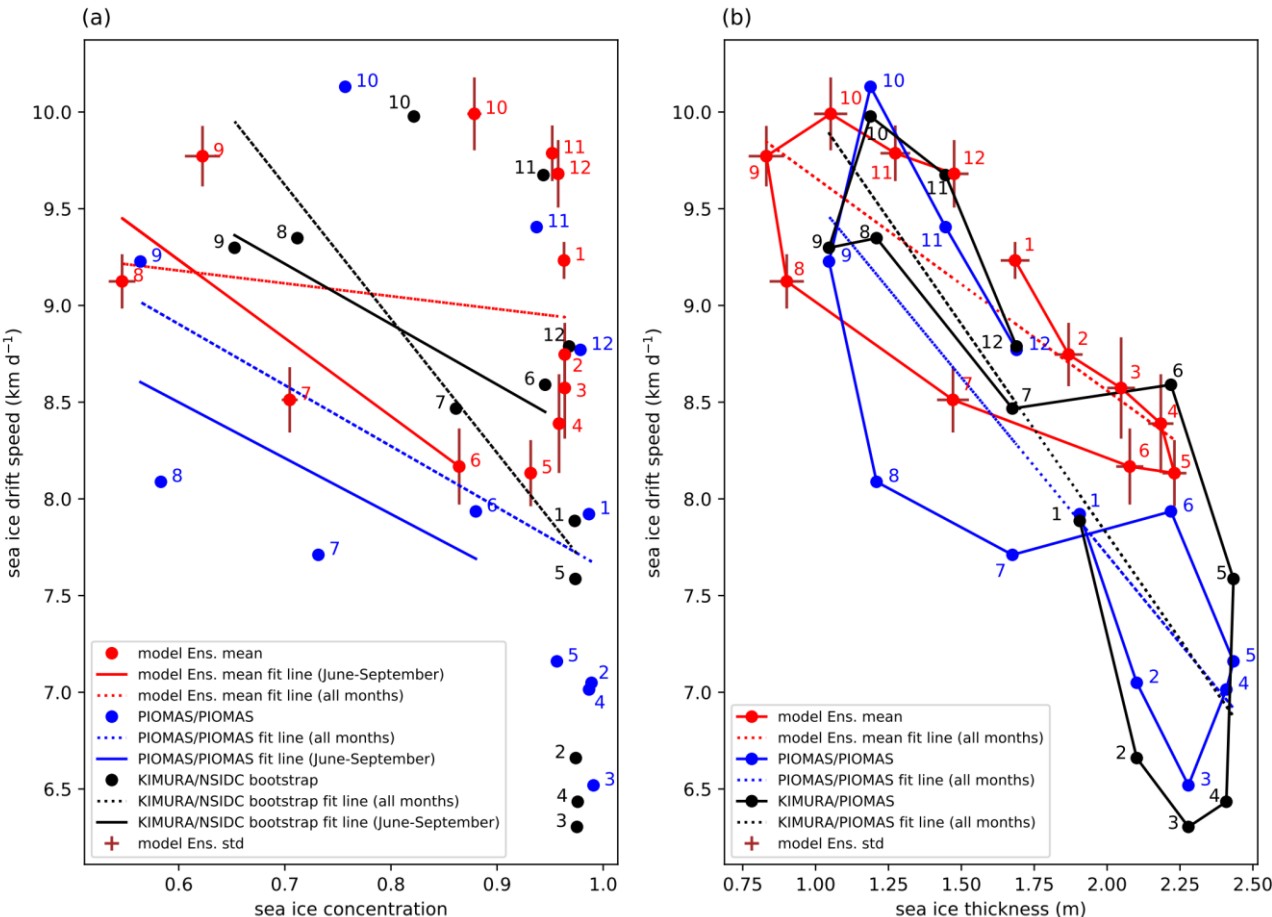

**Figure 6**: Scatter plots of monthly mean sea-ice drift speed against (a) sea-ice concentration and (b) sea-ice thickness, averaged over the period of 2003-2014 and the study domain (indicated in Figure 1). Numbers denote the months. Results are shown for the model (HIRHAM-NAOSIM 2.0) ensemble simulations (red), KIMURA for ice drift speed plus NSIDC bootstrap for ice concentration (black), and PIOMAS data (blue).

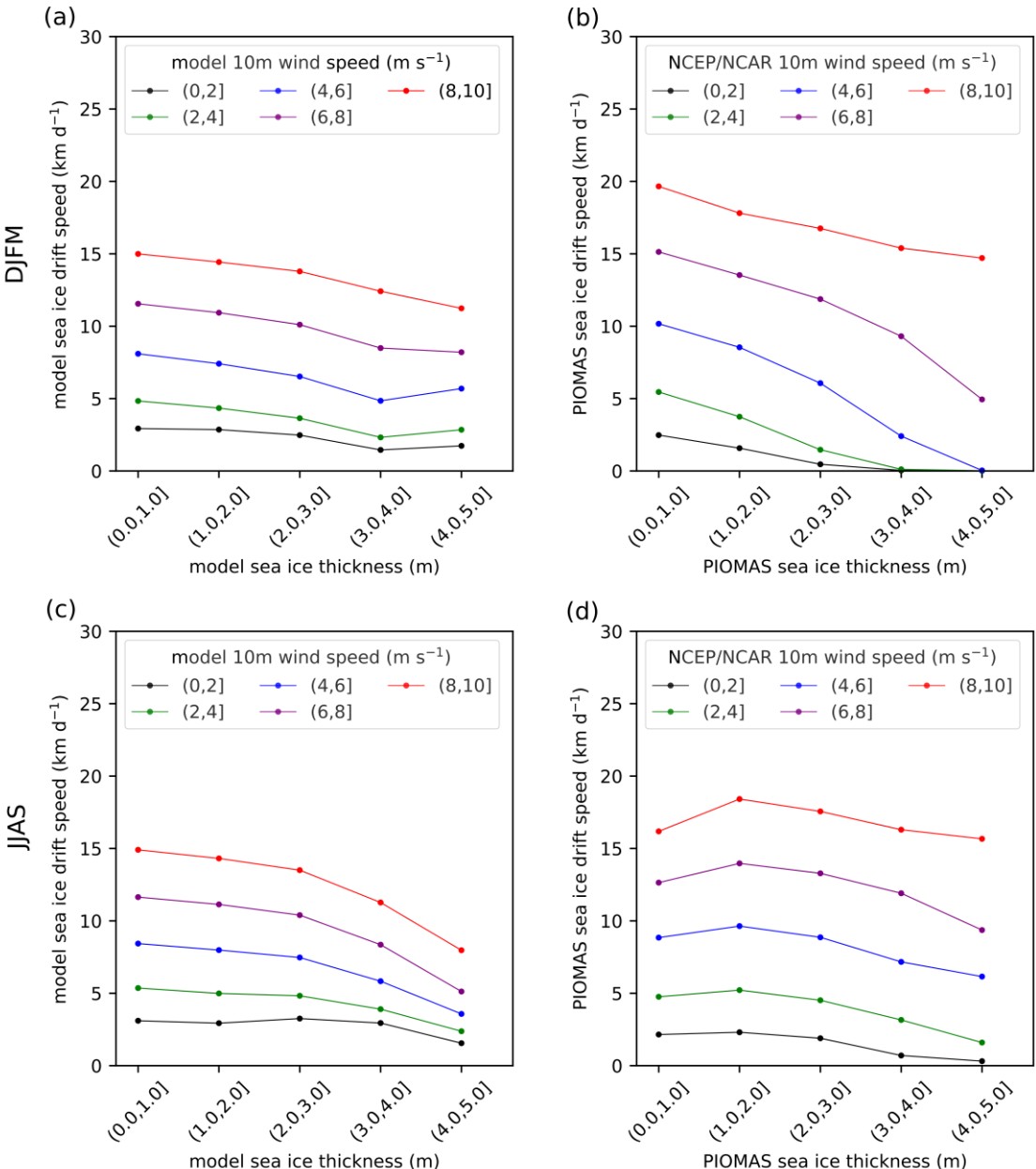

**Figure 7**: Relationship between sea-ice drift speed and sea-ice thickness for different near-surface wind speed classes (different colors) for 2003-2014 (a) winter (DJFM) and (c) summer (JJAS) in the model (HIRHAM-NAOSIM 2.0). (b) and (d) are the same as that for (a) and (c) respectively, but based on PIOMAS data. The points in the plot are the median value of all the daily data and on all grid points within the study domain based on Figure 1. In the labels of different sea-ice thickness and 10-m wind speed classes, "(" means exclusive and "]" means inclusive.



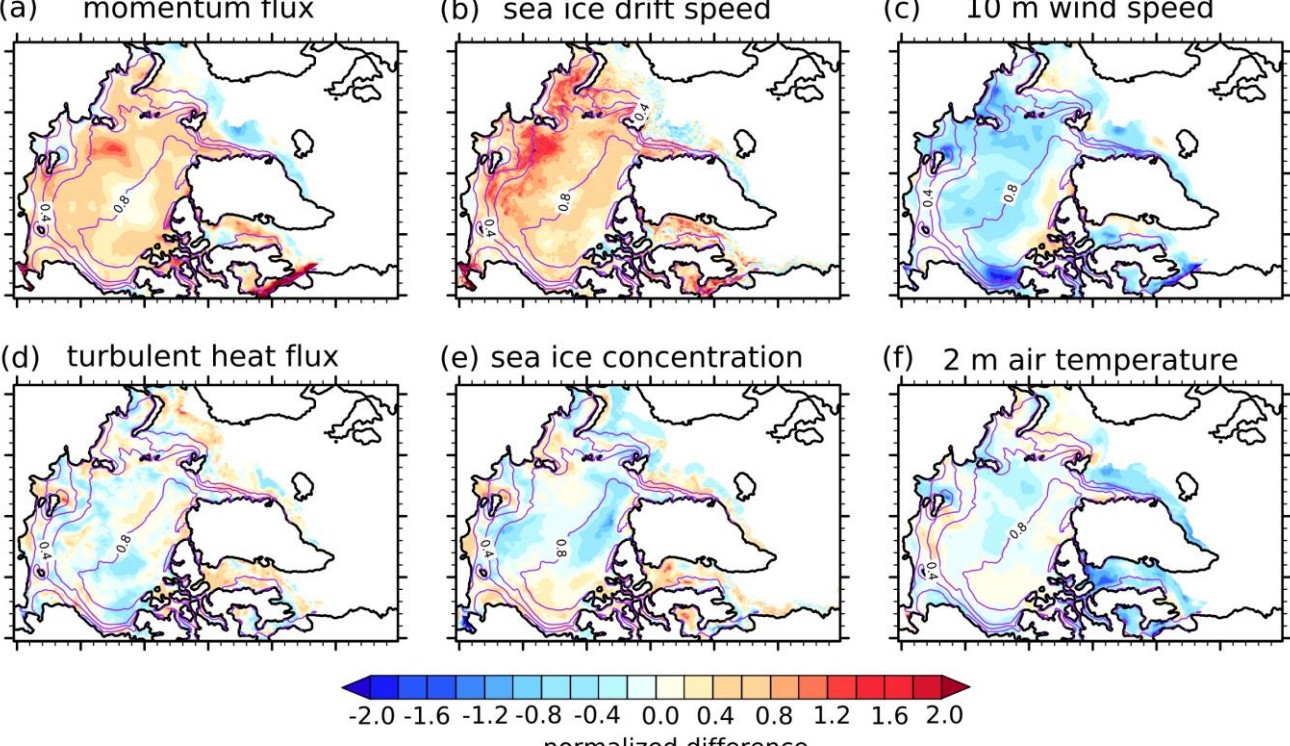

**Figure 8**: Normalized ensemble mean differences (SENS minus CTRL) of (a) momentum flux, (b) sea-ice drift speed, (c) 10-m wind speed, (d) turbulent heat flux, (e) sea-ice concentration, (f) 2-m air temperature for 2007 summer (JJAS). The ensemble mean difference is normalized by the cross-ensemble standard deviation of the differences. The purple contours represent the ensemble mean sea-ice concentration in the CTRL experiment.


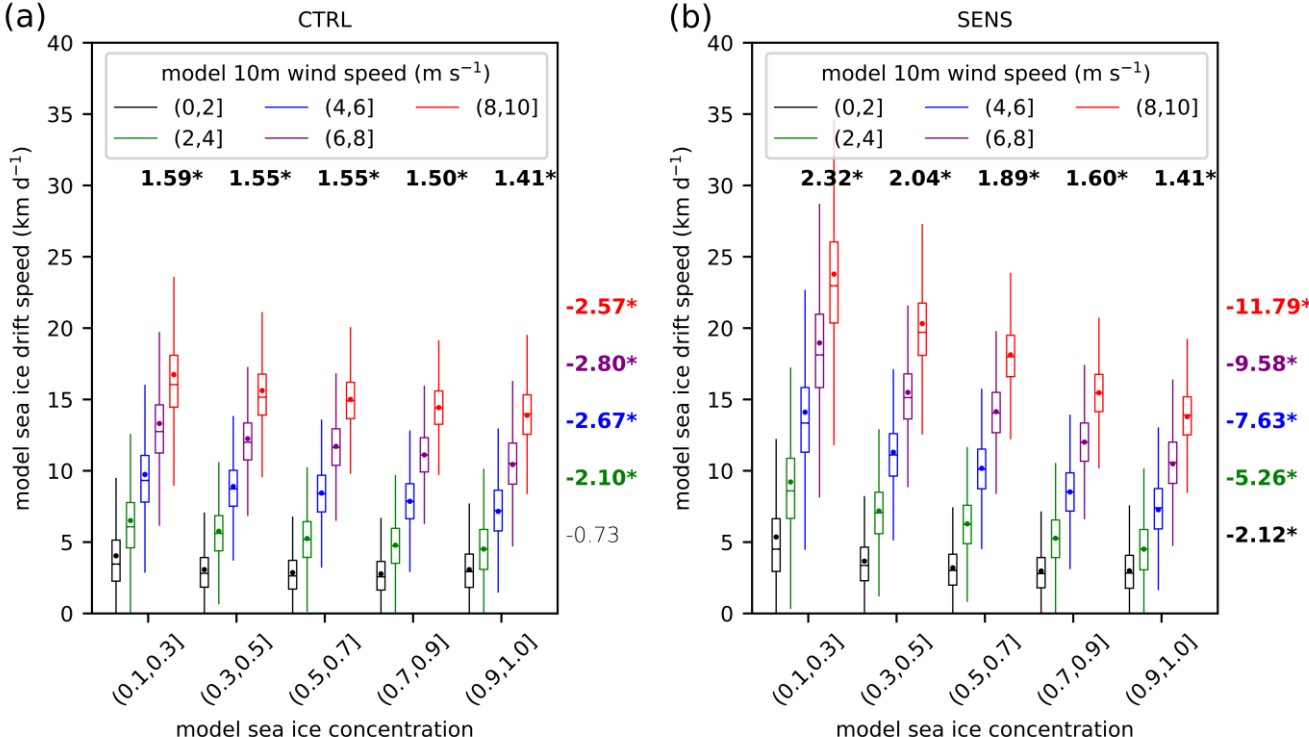


**Figure 9**: Box-whisker plots of the relationship between sea-ice drift speed and sea-ice concentration for different near-surface wind speed classes (different colors) for 2007 summer (JJAS) in (a) CTRL and (b) SENS experiment. The plot is based on daily data and on all grid points within the study domain indicated in Figure 1. The horizontal bar represents the median, the notch represents the 95% confidence interval of the median, the dot represents the mean, the top and bottom of the box represent the 75th and 25th percentiles, the upper/lower whiskers represent the maximum/minimum value within 1.5 times interquartile range (IQR) to 75/25 percentiles. The numbers above the boxplots represent the slopes of near-surface wind and sea-ice drift speed fit lines (unit: km d$^{-1}$ per 1 m s$^{-1}$ wind speed change; font colors as for the wind speed classes). The numbers right of the boxplots represent the slopes of sea-ice concentration and sea-ice drift speed fit lines (unit: km d$^{-1}$ per 10% sea-ice concentration change). A bold and asterisked number indicates that the slope of the fit line is significant at the 95 % level. In the labels of different sea-ice concentration and 10-m wind speed classes, "(" means exclusive and "]" means inclusive.


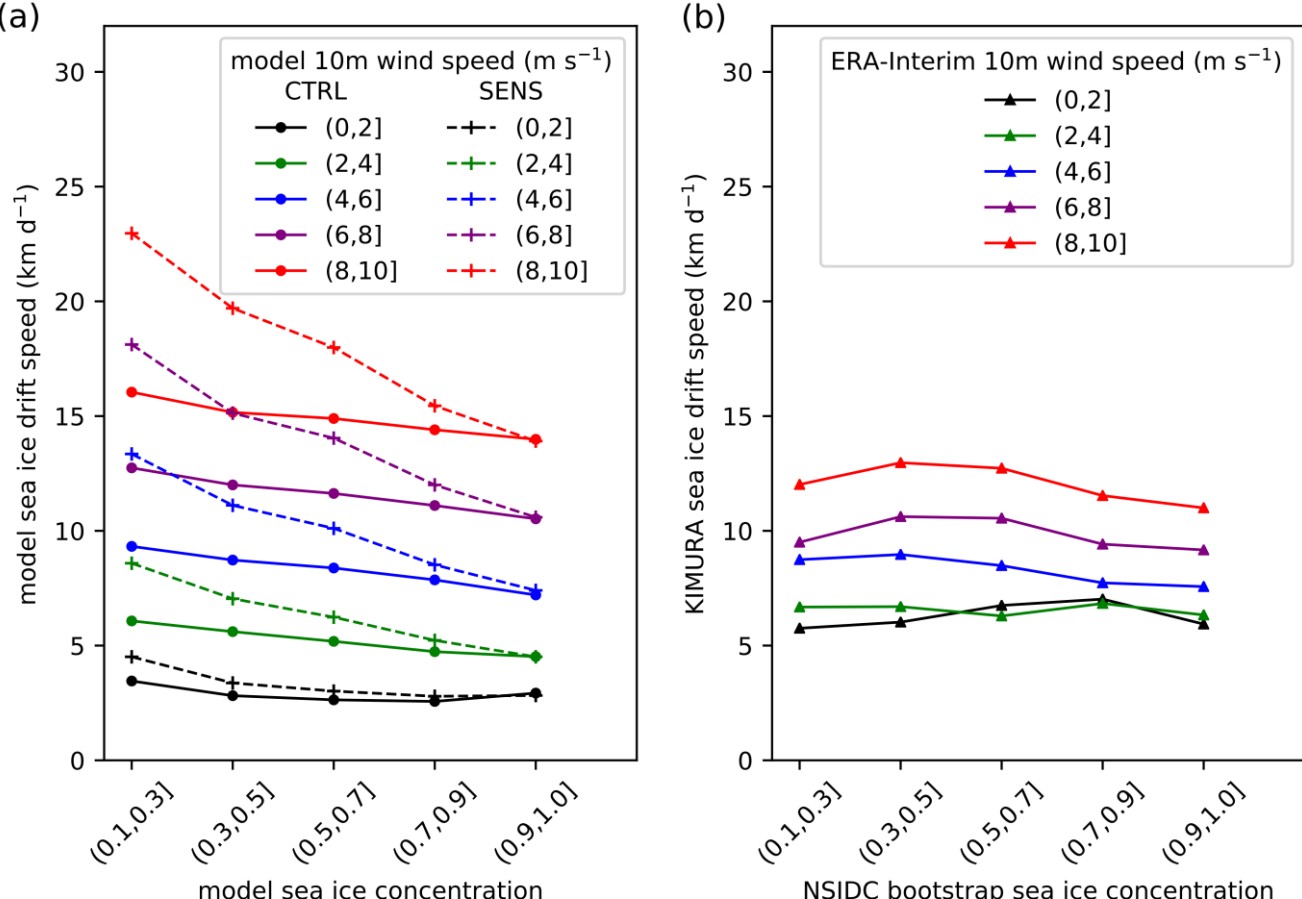

**Figure 10:** The (a) simulated relationship between sea-ice drift speed and sea-ice concentration for different near-surface wind speed classes (different colors) for 2007 summer (JJAS) in CTRL (circle marker and solid line) and SENS (cross marker and dashed line) experiment. The relationship based on KIMUAR sea-ice drift speed, NSIDC bootstrap sea-ice concentration and ERA-interim 10-m wind speed is shown in (b). The points in the plot is the median value of all the daily data and on all grid points within the study domain indicated in Figure 1 under certain wind speed and sea-ice concentration classes. In the labels of different sea-ice concentration and 10-m wind speed classes, "(" means exclusive and "]" means inclusive.


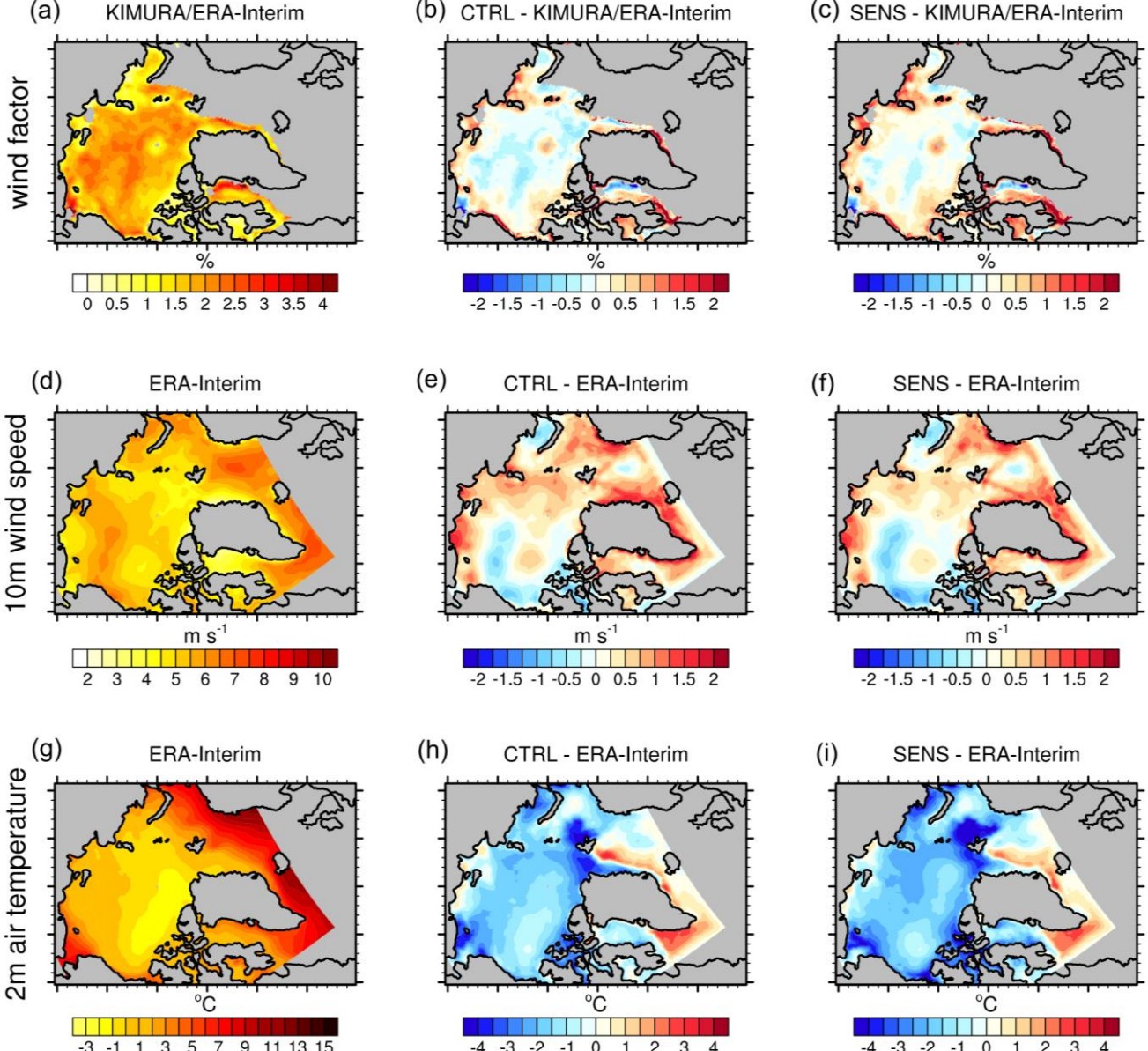

**Figure 11:** The 2007 summer (JJAS) (a) wind factor, (d) 10-m wind speed and (g) 2-m air temperature from ERA-Interim (KIMURA sea-ice drift is used for wind factor calculation). (b) wind factor, (e) 10-m wind speed and (h) 2-m air temperature differences between CTRL experiment and ERA-Interim (KIMURA and ERA-interim for wind factor). (c), (f) and (i) are the same as that for (b), (e) and (h), but show the differences between SENS experiment and ERA-Interim (KIMURA and ERA-interim for wind factor).

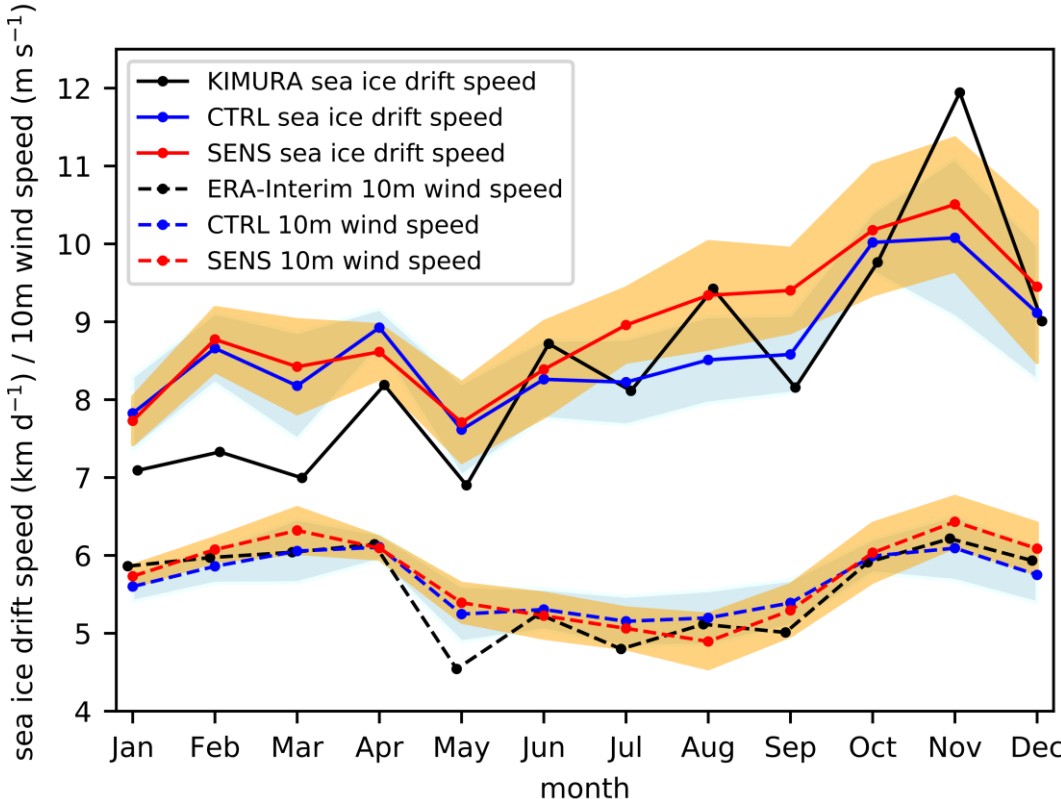

**Figure 12**: Mean annual cycle of sea-ice drift speed [km d$^{-1}$] (solid lines) and 10-m wind speed [m s$^{-1}$] (dashed lines), based on CTRL experiment (ensemble mean; blue lines), SENS experiment (ensemble mean; red lines) and the observation/reanalysis (KIMURA ice drift, ERA-I wind; black lines) for 2007 over the study domain (indicated in Figure 1). The across-ensemble scatters (standard deviation) of the simulations are included as shaded area (light blue for CTRL, orange for SENS).
