# Peer review of "Evaluation of Arctic sea-ice drift and its dependency on near-surface wind and sea-ice concentration and thickness in the coupled regional climate model HIRHAM-NAOSIM"

_The Cryosphere, 2019_

## Referee Comment (RC1) · Anonymous Referee #1 · 25 Sep 2019

**Review of "Evaluation of Arctic sea-ice drift and its dependency on near-surface wind and sea-ice concentration and thickness in the coupled regional climate model HIRHAM-NAOSIM" by Yu, Rinke, Dorn, Spreen, Lüpkes, Sumata and Gryanik.**

This article contains two pieces of work, the first being an assessment of the simulation of sea-ice (in particular sea-ice drift, SID) in a recent ensemble of 10 control members from a coupled regional climate model of the Arctic. The second part is then a sensitivity test, where the parameterization of surface exchange of momentum and heat between the ice and atmosphere is improved based on recent parameterization development documented in other papers. The research is state-of-the-art and an important step forward in the broader aim of trying to improve the fidelity of coupled climate models. The representation of Arctic sea ice is a long-standing known weakness in these models and improving the surface exchange parameterization is tackling one important weakness. The results of this study are mixed. Overall the model CTRL ensemble performs reasonably well compared to observational data sets (although these themselves have deficiencies). The new parameterization acts in a physically realistic way and leads to significant changes in surface variables. However, the authors state that it does not (yet) provide an improved simulation of sea ice, because no model tuning has been carried out yet, and they reserve this for future work.

Overall this is a very commendable study and an important piece of work, so I would like to see it published. I have a number of comments that would improve the manuscript that I'd like to see the authors take on board – some on the presentation that would greatly help new readers and make the work more accessible. The results of the second part of the study seem to end before the punch line! Normally upon introducing a new parameterization, authors invariably tend to find that their new parameterization improves the model. Here we seem to stop short of a full investigation of whether this is the case or not, because some tuning is required. I think this is reasonable, because the paper is already quite long at this point, and I am aware that such tuning is time consuming and opaque. But it does make the paper feel a little unfinished. Have the authors considered making this part 1 of a two-part paper, or at least spelling out in more detail the implied follow up study?

**Specific Comments**

1. The paper's title is long and a bit clumsy (three "ands"). I'd maybe try to reword.

2. In the Introduction I would recommend a short discussion on the quality of the surface exchange parameterization you've introduced. Around L65 you point out that parameterizations without a form drag element for momentum exchange are "poorly constrained" and that a recent observations-based form-drag parameterization has been implemented in a model by Renfrew et al. Here I think you need a few sentences pointing out that the mathematical parameterizations by Lüpkes et al. 2012 and Lüpkes and Gryanik 2015 were constrained by summertime observations over the sea-ice pack (from Andreas et al. 2010) and by limited aircraft observations over the MIZ (marginal ice zone). Then more comprehensively validated and tuned over the MIZ by a larger set of aircraft observations in Elvidge et al. 2016 [Note, this paper is not in the reference list, but there is a citation for Elvidge et al. 2018 in the manuscript, but no reference, so I think you mean the 2016 paper]. Importantly, I think you also need to point out that most of the validation and tuning has been done for momentum exchange (i.e. $C_{DNi}$), very little validation has

been done for heat or moisture exchange (i.e. $C_{HNi}$). The validation and tuning for scalar fluxes is, I think, still something of an open question.

3. Page 4 contains a mathematical description of the new surface exchange parameterization for over sea ice, based on Lüpkes et al. 2012 and Lüpkes and Gryanik 2015. I am familiar with these two papers and I think you are right to leave most of the mathematical details out of this article and refer the reader to these previous articles for details. However, what is tricky is that both of these previous articles are long and technical, with more than 60 and 70 equations in them respectively, and both contain several sets of parameterizations in a hierarchy of complexity. This makes checking the summary you have here difficult, especially as the notation used here is slightly different to the previous papers. I think you need to be more specific and say which equations from the two above papers are implemented and try to use notation that is as close as possible to what is already published (I appreciate this can be difficult). To give one example, equations (3), (4) & (7) all have a '+1', in "$z_{0,i}+1$" – this isn't explained and I don't know what it means. Also equation (8) does not seem to match equation (63) in Lüpkes and Gryanik 2015 – should it? Finally there are a number of parameters set on page 4: $C_{e10}$, $z_{o,f}$, $\beta$, then later on $\alpha$ and $z_{0,i}$. It is not clear where the values for these parameters have come from and I found it difficult to relate them to parameters in the previous studies or in Elvidge et al. 2016. I think this section (2.1.3) could be vastly improved without much additional length or detail. Finally, you don't comment on exchange of moisture, is this changed?

4. In the summary (L440) you state that the SENS simulation is not any better than the CTRL simulations, in terms of sea-ice drift etc. However, you don't really provide evidence for this statement. I think there is evidence in your paper, but you need to discuss it and demonstrate this is the case. Consequently, I would recommend adding another paragraph or two to Section 4, where you discuss the quality of the SENS and CTRL simulations. For example, is it possible to compare the gradients in Fig 6a to Fig 10 and demonstrate whether the CTRL or SENS is better? Could you add some observational data to Fig 10 to show this fact? I appreciate that 'not any better yet' is a bit of a negative result and could be changed by tuning the model, so perhaps you don't want to spend too much time and effort on this aspect. But I think you need to provide a small amount of evidence for this statement.

**Minor Points**

L14 – "…of the Arctic basin" [insert the]
L28 – "sea ice has experienced…"
L66     You categorise Tsamados et al. 2014 as an ice-ocean model, but that study was actually using only a sea-ice model.

L175 – Have you considered a Cryosat product for sea-ice thickness – probably not worth the effort now, but might be interesting for any follow up studies.

L180 – The description of ERAI resolution is misleading. The resolution of the atmospheric model is T255 equivalent to about 80 km resolution, and you have downloaded it on 0.25 degree grid. So please rephrase.

L185 – "resolutions, a bilinear…"
L191 – "as the study…"
L250 – Maybe swap order to winter then summer to match the order earlier in the sentence, i.e. rephrase sentence.

L258 and L262 – Maybe rephrase to state 'in winter…" and "in summer, …" clearly at the beginning of the statement, rather than hidden in the middle of the sentence.

L335-340 – Can you cite some evidence that PIOMAS is wrong here – I think it is incorrect and it is certainly inconsistent with the model.

L345 – You don't discuss Fig 6a at all. Is it needed? Perhaps it should be discussed later.

**Figures**

**Figures 1, 3, 8, S1, S2, S3, S4**
These all use the same colormap which is a blue-white-red (diverging colormap). Such colormaps are ideal for difference plots, e.g. Fig 1b,d, but are an odd choice for non-diverging fields, such as Fig 1a,c. I wonder if you are better changing colormap for the left-hand columns in all of these plots.

**Figure 4**
Unfortunately, this is really hard to read at this size (printed A4). I also think it has too much unnecessary detail in it. You have 10 wind speed classes. Do you really need this many classes? I think you'd get the same result with 2 m/s bins and it would be much clearer. Also do you include winds >10 m/s in the (9,10] bin? Note I have taken (1,2] to mean winds between 1 and 2 (inclusive) m/s – you should explain this in first caption.

Secondly you have 9 SIC bins – again this is a lot and there seems very little difference in the results between adjacent bins. I'd perhaps recommend fewer bins, perhaps (0,0.1], (0.1,0.3], (0.3,0.5], (0.5,0.7], (0.7,0.9], (0.9,1.0]. This keeps the 'end' bins separate as these are more interesting. At present this Fig 4 and also 9 has so much detail and numbers, that the main message is a bit hidden.

Finally, it may be worth noting how much data is in these bins. Although the bins are the same size (0.2 in ice fraction for example), the distributions mean there could be relatively few points in some bins.

**Figure 5**
Same comments as above really and note some of the colours are very faint (7,8) class. These plots are more readable but need to be consistent with Fig 4.

**Missing Reference:**
Elvidge, A.D., I.A. Renfrew, A.I. Weiss, I.M. Brooks, T.A. Lachlan-Cope, and J.C. King 2016: Observations of surface momentum exchange over the marginal-ice-zone and recommendations for its parameterization, *Atmospheric Chemistry and Physics,* **16,** 1545-1563. doi:10.5194/acp-16-1545-2016

---

## Referee Comment (RC2) · Anonymous Referee #2 · 20 Oct 2019

General description The Manuscript validates a fully coupled ocean, sea ice and atmospheric circulation model (HIRHAM-NAOSIM). Main focus is on the correlation between the sea-ice drift, sea-ice conditions and the near surface wind speed. The model is validated towards remotely sensed data and another model (PIOMAS). The latter is more a comparison than a validation.

Comparisons has been made at two time scales. First the seasonal variation has been compared, then the daily variations and correlations between mainly ice drift and wind speed in different sea ice regimes.

Results are in general at the level of other model systems, however a few pointers are provided to places where HIRHAM-NAOSIM performs better than PIOMAS

Introduction of sea-ice form drag influences the drift speed but this does not improve the overall performance.

Mayor revisions/concerns

A paper that validates a model is of relevance, however it seems like there are many references to the 2019 paper by Dorn et al. Without having read this I am a little puzzled whether this manuscript is more of the same of if it points to new findings. Especially when tuning of the form drag is postponed to a later paper.

My main concern with this manuscript is that it presents many numbers and correlations but there is a lack of introduction, perspective and discussion. A few lines is mentioned in the end of section 4 where ocean forcing is mentioned. I think that this should be the start of a discussion that discuss the reasons why for instance the seasonal cycle is poorly represented. How well is the internal ice pressure described? Are observations always the truth? For instance what are the uncertainties/biases of the KIMURA dataset.

The article points to the lack of a seasonal cycle and a bad timing of the minimum. My opinion would be that the minimum is more a matter of lack of a seasonal cycle and that this is a random minimum that is irrelevant as long as the seasonal cycle is not present.

From line 52 and the next few lines a method for validation is mentioned. I would recommend to move this into section 2 and describe what this validation method do.

Some of the findings are close related. Higher ice drift will lead to lower ice thickness and again higher ice drift. Therefore a comparison with for instance PIOMAS tells you more about the current state of the model than a direct bias (at least that would be my opinion). The comparisons are valid but I will be hesitant to say that for instance the

internal strength of the model is too weak. A relevant discussion related to PIOMAS would be to discuss the difference between a forced ocean-sea ice model and a fully coupled model ocean-sea ice-atm model. Are there features that could be described by this?

Minor corrections Line 44: In my opinion the comparison of CMIP 3 models is outdated. The reference provided afterwards is more relevant (Tandon et al 2018).

Line 50: Please don't start the section with Thus. For instance change to: This paper/manuscript has two aims. Line 54. Stating that an observation is rare seems a bit short and subjective. They do exist (RGPS buoys, SAR drift), however these are not present for the entire period. Choosing not to use them is valid but again a few more lines on why would be nice.

Line 75: Replace with: The organization of this paper is as follows: Section 2.

Line 84 to 95: A map of the domain and the where the boundaries extend to would improve the understanding of the model domain.

Line 92 reference a dynamic-thermodynamic model described by Harder is an upgrade? What is upgraded. Dynamics are referenced to 1979 and thermodynamics to 1976. Maybe "update" should be removed or explicitly explained what is the update.

Line 104: How is the spinup designed? Running one year 22 times? Has the model bin spun up properly or is the ensemble a representation of the spinup? A bit more elaboration of the choices would be nice. Is Levitus data near the area of interest good enough? Does this imply that the variations seen only originates from the atmosphere?

Line 157: Validation towards AMSRE. Is the ice drift It would be interesting to see how the model performed vs RGPS buoys and Sentinel 1 SAR icedrift data. Alternatively an evaluation of the uncertainty of the chosen drift product versus the bias/uncertainty of the model results.

Line 162: As partly mentioned the comparison with PIOMAS just shows whether

NAOSIM provides the same as PIOMAS. Why not use Icesat as mentioned in the discussion about PIOMAS. Admitted there are relatively high uncertaintes on ice thickness products like IceSat, however reference a model and motivate this choice by its skill vs another product seems weird. Other data sets that can be used are operation ice bridge and Cryosat. They do not cover the full period and domain but they can do as Ground Truth.

Line 187 - 189. Is there a reason for excluding spring and fall?.

Line 211-216 Not sure why it is required to include such a long description of why sea ice drift is influenced by thickness, concentration and wind speed. This is stated in several articles. Just state that the drift is governed mainly by ice conditions, wind speed and ocean currents (less important).

Line 240 - Small variation of wind don't explain variation of ice drift. The modelled ice drift seems to be controlled mostly by the wind, however this is in contrast to obs.

245 - 250 Again too high correlated wind and ice drift speed in winter. Other factors/forcing of the dynamics of the sea-ice must impact. A discussion of these would be relevant in a discussion section.

Line 260. I thought that there is a dynamical forcing between ocean and sea ice everywhere. This should be more specific.

Line 300 what is the method? Short description please. Same reference is made in introduction

Line 350 Abrupt end to line.

Figure 4 and 5 are hard to read. Please increase font size

---

## Author Comment (AC1) · 10 Dec 2019

**Answer to the comments of Referee #1**

We would like to thank Referee #1 for his/her suggestions to improve our paper. All comments have been addressed and a point by point response is provided below each comment. The reviewer comments are written in black, our answer in blue and the corrections in the paper are highlighted in red. The line numbers, which are used in the answers, correspond to the new version of the manuscript (PDF file) unless otherwise indicated.

**General comments**

This article contains two pieces of work, the first being an assessment of the simulation of sea-ice (in particular sea-ice drift, SID) in a recent ensemble of 10 control members from a coupled regional climate model of the Arctic. The second part is then a sensitivity test, where the parameterization of surface exchange of momentum and heat between the ice and atmosphere is improved based on recent parameterization development documented in other papers. The research is state-of-the-art and an important step forward in the broader aim of trying to improve the fidelity of coupled climate models. The representation of Arctic sea ice is a long-standing known weakness in these models and improving the surface exchange parameterization is tackling one important weakness. The results of this study are mixed. Overall the model CTRL ensemble performs reasonably well compared to observational data sets (although these themselves have deficiencies). The new parameterization acts in a physically realistic way and leads to significant changes in surface variables. However, the authors state that it does not (yet) provide an improved simulation of sea ice, because no model tuning has been carried out yet, and they reserve this for future work.

Overall this is a very commendable study and an important piece of work, so I would like to see it published. I have a number of comments that would improve the manuscript that I'd like to see the authors take on board – some on the presentation that would greatly help new readers and make the work more accessible. The results of the second part of the study seem to end before the punch line! Normally upon introducing a new parameterization, authors invariably tend to find that their new parameterization improves the model. Here we seem to stop short of a full investigation of whether this is the case or not, because some tuning is required. I think this is reasonable, because the paper is already quite long at this point, and I am aware that such tuning is time consuming and opaque. But it does make the paper feel a little unfinished. Have the authors considered making this part 1 of a two-part paper, or at least spelling out in more detail the implied follow up study?

We added a new paragraph to describe the ideas for the implied follow-up study (line 514):

"Although the new parameterization does not improve the simulated SID dependency on WS and sea-ice conditions compared to observations/reanalysis, the sensitivity study clearly shows that the new parameterization does increase the SID due to the added form drag. In a follow-up study, we are going to put efforts therefore on several aspects to improve the simulations. First, tunable parameters of the new parameterization, such as $z_0$, $z_t$, $Ce_{10,i}$ , $Ce_{10,k}$ and $\beta$ represent an opportunity to better adapt the form drag parameterization itself to the observations. A first step could be the use of values found by Elvidge et al. (2016).  A large effect can be expected by a modification of the skin drag coefficient, since a large region would be affected, and large variations in the drag due to pressure ridges allow a wide range of values. Second, model parameters outside the new parameterization, which have direct impact on SID, like ice strength and ocean-ice drag coefficient, need to be harmonized with the new parameterization, since their values were chosen empirically in terms of adequately balanced performance of the model. A key is probably the oceanic form drag. Its effect is accounted for in the present study only indirectly via the constant oceanic drag coefficient. Such a parametrization is probably too simple, especially when atmospheric form drag is included (see also Tsamados et al., 2014). Birnbaum (2002) as well as Lüpkes et al. (2012b) found in a mesoscale modelling study that oceanic form drag can have a strong decelerating effect on SID especially when the sea ice concentration is low so that the discussed drawbacks for small sea ice fraction would be reduced or even removed. This effect of form drag on SID was discussed also by Steele et al. (1989). The parametrizations are evidently not balanced anymore after improving one key process of the SID-related atmosphere-ocean-ice interaction. A previous study on the surface-albedo feedback by Dorn et al. (2009) showed that an improved simulation can only be achieved by a harmonized combination of more sophisticated parameterizations of the related sub-processes. It can be assumed that this holds true for the SID-related sub-processes."

Birnbaum, G., and Lüpkes, C.: A new parameterization of surface drag in the marginal sea ice zone, Tellus A: Dynamic Meteorology and Oceanography, 54, 107-123, 10.3402/tellusa.v54i1.12121, 2002.

Steele M., Morison, J.H., Untersteiner N. (1989) The partition of air-ice ocean momedntum exchange as a function of sea ice concentration, floe size, and draft. J. Geophys. Res. 94: 12739-12750.

We also revised the discussion of the new parametrization at line 572:

"The inclusion of the melt pond effect on form drag in the model might be beneficial. In the current version, form drag was only considered at the edges of ice floes, mainly in the marginal sea-ice zone, but not on top of the ice, where melt ponds cause form drag also during summer (Andreas et al., 2010; Lüpkes et al., 2012a). Additional form drag at the ice-ocean interface may further improve the simulated SID-WS relation, because the oceanic form drag has normally the opposite effect on the ice motion as the

atmospheric form drag (Steele et al., 1989; Lüpkes et al., 2012b). Systematic biases in the reanalysis used for the calculation of the 'observed' wind factor cannot be excluded, since form drag is not taken into account in the underlying atmospheric model. Therefore, the increased deviation of the simulated SID-WS relation from the observations/reanalysis does not necessarily mean that the implemented new parameterization worsens the SID-WS relation."

**Specific Comments**

1. The paper's title is long and a bit clumsy (three "ands"). I'd maybe try to reword.

We agree and modified the title to

"Evaluation of Arctic sea-ice drift and its dependency on near-surface wind and sea-ice conditions in the coupled regional climate model HIRHAM-NAOSIM"

2. In the Introduction I would recommend a short discussion on the quality of the surface exchange parameterization you've introduced. Around L65 you point out that parameterizations without a form drag element for momentum exchange are "poorly constrained" and that a recent observations-based form-drag parameterization has been implemented in a model by Renfrew et al. Here I think you need a few sentences pointing out that the mathematical parameterizations by Lüpkes et al. 2012 and Lüpkes and Gryanik 2015 were constrained by summertime observations over the sea-ice pack (from Andreas et al. 2010) and by limited aircraft observations over the MIZ (marginal ice zone). Then more comprehensively validated and tuned over the MIZ by a larger set of aircraft observations in Elvidge et al. 2016 [Note, this paper is not in the reference list, but there is a citation for Elvidge et al. 2018 in the manuscript, but no reference, so I think you mean the 2016 paper]. Importantly, I think you also need to point out that most of the validation and tuning has been done for momentum exchange (i.e. $C_{DNi}$), very little validation has been done for heat or moisture exchange (i.e. $C_{HNi}$). The validation and tuning for scalar fluxes is, I think, still something of an open question.

We corrected the citation and added a short discussion on the quality of the surface exchange parameterization following the Referee's suggestions (line 85):

"The mathematical parameterizations proposed by Lüpkes & Gryanik (2015) were constrained by summertime observations over the sea-ice pack and by aircraft observations over the MIZ (marginal ice zone) during winterly conditions. Later the parameterizations were once more validated using a larger and independent set of aircraft data obtained from campaigns during different seasons (Elvidge et al., 2016). This validation work concerned the momentum transport, but the assumptions of Lüpkes & Gryanik (2015) about heat and moisture flux over the MIZ could not yet be evaluated by measurements. Thus, further research is necessary on this issue."

3. Page 4 contains a mathematical description of the new surface exchange parameterization for over sea ice, based on Lüpkes et al. 2012 and Lüpkes and Gryanik 2015. I am familiar with these two papers and I think you are right to leave most of the mathematical details out of this article and refer the reader to these previous articles for details. However, what is tricky is that both of these previous articles are long and technical, with more than 60 and 70 equations in them respectively, and both contain several sets of parameterizations in a hierarchy of complexity. This makes checking the summary you have here difficult, especially as the notation used here is slightly different to the previous papers. I think you need to be more specific and say which equations from the two above papers are implemented and try to use notation that is as close as possible to what is already published (I appreciate this can be difficult). To give one example, equations (3), (4) & (7) all have a '+1', in "$z_{0,i}+1$" – this isn't explained and I don't know what it means. Also equation (8) does not seem to match equation (63) in Lüpkes and Gryanik 2015 – should it? Finally there are a number of parameters set on page 4: $C_{e10}$, $z_{0,f}$, b, then later on a and $z_{0,i}$. It is not clear where the values for these parameters have come from and I found it difficult to relate them to parameters in the previous studies or in Elvidge et al. 2016. I think this section (2.1.3) could be vastly improved without much additional length or detail. Finally, you don't comment on exchange of moisture, is this changed?

We added the source of equations (1) to (4) and the explanation of adding '+1' in equations (3) and (4) at line 158:

"Equations (1) to (4) are common descriptions of air-ice momentum and heat transfer coefficients except that '+1' was added to both $z_L/z_{0,i}$ and $z_L/z_{t,i}$ in equations (3) and (4). This is done in the model to avoid that the argument of the logarithm can go to zero, for which $C_{d,i}$ ($C_{h,i}$) would go to infinity (see also Giorgetta et al., 2013)."

Giorgetta, M. A., Roeckner, E., Mauritsen, T., Bader, J., Crueger, T., Esch, M., Rast, S., Kornblueh, L., Schmidt, H., and Kinne, S.: The atmospheric general circulation model ECHAM6-model description, 2013.

The source of equations (5) to (7) were added at line 169:

"Equations (5) is obtained by combining the equation (6), (52) and (70) of Lüpkes & Gryanik (2015). Equations (6) is obtained by combining the equation (9), (64) and (74) of Lüpkes & Gryanik (2015). After adding '+1' both to $10/z_0$ and $z_L/z_0$ and replacing $z_0$ with $z_{0,f}$ in equation (65) of Lüpkes and Gryanik (2015), $C_{dn,f}$ is calculated as

$$C_{dn,f} = C_{e10} \left[\frac{\ln(10/z_{0,f}+1)}{\ln(z_L/z_{0,f}+1)}\right]^2 A(1-A)^\beta \qquad (7)"$$

Equation (8) represents a simple algebraic transformation of equation (60) by Lüpkes and Gryanik (2015) making use of their equations (59) and (61). We added one sentence to clarify how we got the equation (8) at line 184:

"Equation (8) represents a simple algebraic transformation of equation (60) by Lüpkes and Gryanik (2015) making use of their equations (59) and (61) with $\alpha_f = \alpha$."

We added the source of the values for $C_{e10}$ , $z_{0,f}$ and $\beta$ at line 176:

"The value of $C_{e10}$ is the average given in equations (48) and (49) by Lüpkes and Gryanik (2015). The value of $z_{0,f}$ is an average resulting from measured roughness lengths by various campaigns considered by Andreas et al. (2010), Lüpkes et al. (2012a) and Castellani et al. (2014). Note that this value is not critical for the parametrization. The value of $\beta$ comes from equation (59) by Lüpkes et al. (2012a)."

The exchange of moisture is treated in the same way as the exchange of heat, meaning that the same coefficients are used.

4. In the summary (L440) you state that the SENS simulation is not any better than the CTRL simulations, in terms of sea-ice drift etc. However, you don't really provide evidence for this statement. I think there is evidence in your paper, but you need to discuss it and demonstrate this is the case. Consequently, I would recommend adding another paragraph or two to Section 4, where you discuss the quality of the SENS and CTRL simulations. For example, is it possible to compare the gradients in Fig 6a to Fig 10 and demonstrate whether the CTRL or SENS is better? Could you add some observational data to Fig 10 to show this fact? I appreciate that 'not any better yet' is a bit of a negative result and could be changed by tuning the model, so perhaps you don't want to spend too much time and effort on this aspect. But I think you need to provide a small amount of evidence for this statement.

We agree that it is helpful to support the statement that SENS is not better than CTRL in term of SID and SID-WS relation. We followed the Referee's suggestions and modified the original Figure 10 to include the boxplot of sea-ice drift speed against different sea-ice concentration and wind speed from observation and reanalysis. We added a new Figure 11 that compares wind factor, 10-m wind speed and 2-m air temperature from observation/reanalysis and from the CTRL and SENS simulations for summer 2007. We also added a new Figure 12 that compares the seasonal cycle of Arctic basin-wide averaged sea-ice drift speed and 10-m wind speed from observation/reanalysis and from CTRL and SENS. The modified and new figures show that the new parameterization both slightly reduces and increases the wind factor and 10-m wind bias over the Arctic dependent on location. The discussions based on these figures are added as a new subsection "4.2 Model versus observation":

**"4.2 Model versus observation**

The increased SID dependency on WS and SIC in SENS compared to CTRL does not reduce the deviation to observation/reanalysis. In contrast, Figure 10 shows that the overestimation of SID dependency on WS and SIC in SENS is larger than in CTRL.

Figure 11 shows the spatial distribution of the summer 2007 wind factor, WS and near-surface air temperature from observation/reanalysis data, and the deviations of these three variables in CTRL and SENS from observation/reanalysis. It is obvious that both the bias patterns and magnitudes of CTRL and SENS are quite similar. Considering the ensemble mean bias and taking the internal model variability into account, it is hard to detect significant changes in SENS, compared to CTRL, as discussed above (Figure 8).

Although the ensemble mean of Arctic basin-wide mean SID from July to September in SENS is larger than in CTRL, the differences are not statistically significant due to a large ensemble spread (Figure 12). Actually, there are no statistically significant differences in the Arctic basin-wide mean SID between CTRL and SENS in all months. From January to May, the simulated Arctic basin-wide mean SID (both in CTRL and SENS) are higher than that in KIMURA. With respect to the summer months (June to September), the August simulated Arctic basin-wide mean SID in CTRL is lower than in KIMURA, while the July and September simulated Arctic basin-wide mean SID in SENS are higher than in KIMURA. For the Arctic basin-wide mean WS, there is no significant difference between CTRL and SENS as well as between model and reanalysis, except for May, when both model simulations significantly overestimate the WS."

The modified Figure 10 is as follow:

[Figure]

**Figure 10:** The (a) simulated relationship between sea-ice drift speed and sea-ice concentration for different near-surface wind speed classes (different colors) for 2007 summer (JJAS) for CTRL (circle marker and solid line) and SENS (cross marker and dashed line) experiment. The relationship based on KIMUAR sea-ice drift speed, NSIDC bootstrap sea-ice concentration and ERA-interim 10-m wind speed is shown in (b). The points in the plot is the median value of all the daily data and on all grid points within the study domain indicated in Figure 1 under certain wind speed and sea-ice concentration classes.

The added Figure 11 and 12 are as follows:

[Figure]

**Figure 11:** The 2007 summer (JJAS) wind factor, 10-m wind speed and 2-m air temperature from ERA-Interim (KIMURA sea-ice drift is used for wind factor calculation) and the deviations of these three variables in the ensemble mean of CTRL and SENS experiments from ERA-Interim (KIMURA and ERA-interim for wind factor).

[Figure]

**Figure 12**: Mean annual cycle of sea-ice drift speed [km d$^{-1}$] (solid lines) and 10-m wind speed [m s$^{-1}$] (dashed lines), based on CTRL experiment (ensemble mean; blue lines), SENS experiment (ensemble mean; red lines) and observation/reanalysis (KIMURA ice drift, ERA-I wind; black lines) for 2007 over the study domain (indicated in Figure 1). The across-ensemble scatter (standard deviation) of the simulations is included as shaded area (light blue for CTRL, orange for SENS).

**Minor Points**

L14 – "…of the Arctic basin" [insert the]

Changed as suggested.

L28 – "sea ice has experienced…"

Changed as suggested.

L66 You categorise Tsamados et al. 2014 as an ice-ocean model, but that study was actually using only a sea-ice model.

We agree and modified the introduction of the study of Tsamados et al. (2014) and other studies that also include sea-ice form drag in the model simulation as follows (line 67):

"Several model studies that include the sea-ice form drag were carried out (Castellani et al., 2018; Renfrew et al., 2019; Tsamados et al., 2014). Tsamados et al. (2014) implemented a complex sea-ice form drag parameterization based on many sea-ice cover properties (e.g. sea-ice concentration, vertical extent and area of ridges, freeboard and floe draft, and the size of floes and melt ponds) into the stand-alone sea-ice model

CICE. Castellani et al. (2018) implemented a simpler sea-ice form drag parameterization that only relies on sea-ice deformation energy and concentration into the coupled ocean-sea ice model MITgcm. Both studies showed improvement in sea-ice drift after the form drag had been included. Recently, Renfrew et al. (2019) implemented an observation based parameterization of atmospheric form drag caused by floe edges based on Lüpkes et al., (2012a), Lüpkes & Gryanik (2015) and Elvidge et al., (2016) into a stand-alone atmosphere model. The simulation results show an improved agreement of mean atmospheric variables and turbulent fluxes with measurements in cold-air outbreak situations over the Fram Strait when form drag is included."

L175 – Have you considered a Cryosat product for sea-ice thickness – probably not worth the effort now, but might be interesting for any follow up studies.

We have considered to use Cryosat2, but Cryosat2 is only available from 2010 onwards and from October to next April. Therefore, Cryosat2 does not cover the whole period of 2003-2014 and does not provide the summer data. Nevertheless, we decided to add the comparison of sea-ice thickness from Cryosat2 and from the model simulations during winter 2010-2014 in supplementary Figure S2. It shows that the sea-ice thickness difference between Cryosat2 and the model is qualitatively similar to the difference between PIOMAS and the model.

We added according sentence in section 3.1:

"Analysis of the SIT differences between HIRHAM-NAOSIM and CryoSat2 during winter 2010-2014 (Figure S2) confirms that HIRHAM-NAOSIM underestimates the SIT over the central Arctic and north of the Canada Archipelago and Greenland, at least in winter."

L180 – The description of ERAI resolution is misleading. The resolution of the atmospheric model is T255 equivalent to about 80 km resolution, and you have downloaded it on 0.25 degree grid. So please rephrase.

We added more information to the description of the ERA-Interim data at line 228:

"For the near-surface wind speed (WS), daily 10-m wind speed from ERA-I is used. The ERA-I data were downloaded from the MARS archive at ECMWF and interpolated to the same 0.25º x 0.25º grid as used in the model's atmosphere component HIRHAM5."

L185 – "resolutions, a bilinear…"

Changed as suggested.

L191 – "as the study…"

Changed as suggested.

L250 – Maybe swap order to winter then summer to match the order earlier in the sentence, i.e. rephrase sentence.

The sentence was rephrased as suggested (line 322):

"Averaged over the study domain, the simulated wind factor is 1.77% in winter and 1.87% in summer, which agrees with the observations/reanalysis (KIMURA ice drift/ERA-I wind) in the sense that the averaged wind factor is smaller in winter (1.42%) than in summer (1.96%)."

L258 and L262 – Maybe rephrase to state 'in winter…" and "in summer, …" clearly at the beginning of the statement, rather than hidden in the middle of the sentence.

The sentences were rephrased as suggested (line 333):

"In winter, however, the simulated wind factor is overestimated compared to the KIMURA/ERA-I data almost everywhere over the study domain, with the maximum bias reaching 1% over the thick ice north of the Canadian Archipelago (Figure 3). In summer, the modelled wind factor peaks (~3%) along the marginal ice zone, such as in the coastal Beaufort Sea.

…and (line 340):

In contrast to winter, the modelled wind factor in summer is underestimated over the study domain."

L335-340 – Can you cite some evidence that PIOMAS is wrong here – I think it is incorrect and it is certainly inconsistent with the model.

We don't have a reference yet, but we think one possible explanation that PIOMAS gives a SID-SIC relation inconsistent with the observed relation is a violation of physical consistency in the modeling system by the data assimilation. We added the following sentences in paragraph 3 of section 3.3.2 to elaborate our explanation:

"PIOMAS gives a SID-SIC relation that is inconsistent with the observed relation. Thus, the PIOMAS relation might violate physical consistency due to the used assimilation method as explained in the following. PIOMAS employs the optimal interpolation method to obtain a realistic sea ice field (concentration). This procedure contains addition/subtraction of sea ice into the system at every assimilation time step, when the modeled sea ice concentration differs from the observed one. Due to the addition/subtraction of sea ice (called increment or innovation in the terminology of

data assimilation), PIOMAS does not necessarily preserve the physical relations described in the underlying sea ice-ocean model. Such an inconsistency is one of the drawbacks of the optimal interpolation method and therefore relations between assimilated physical properties should be examined with caution."

L345 – You don't discuss Fig 6a at all. Is it needed? Perhaps it should be discussed later.

Actually, we already discuss Figure 6a at the beginning of Section 3.3.1. We agree that it is easy to overlook Figure 6a or 6b because they are not discussed together. We start Section 3.3.1 now with the sentence 'Figure 6a shows…'

**Figures**

**Figures 1, 3, 8, S1, S2, S3, S4**
These all use the same colormap which is a blue-white-red (diverging colormap). Such colormaps are ideal for difference plots, e.g. Fig 1b,d, but are an odd choice for non-diverging fields, such as Fig 1a,c. I wonder if you are better changing colormap for the left-hand columns in all of these plots.

We understand the reviewer's concern about using blue-white-red color map for non-diverging fields. Therefore, we replaced the blue-white-red color map in Figures 1, 3, S1, S2, S3 and S4 with a yellow-red color map for all non-diverging fields. The color map in Figure 8 was not changed because there is no non-diverging field.

**Figure 4**
Unfortunately, this is really hard to read at this size (printed A4). I also think it has too much unnecessary detail in it. You have 10 wind speed classes. Do you really need this many classes? I think you'd get the same result with 2 m/s bins and it would be much clearer. Also do you include winds >10 m/s in the (9,10] bin? Note I have taken (1,2] to mean winds between 1 and 2 (inclusive) m/s – you should explain this in first caption. Secondly you have 9 SIC bins – again this is a lot and there seems very little difference in the results between adjacent bins. I'd perhaps recommend fewer bins, perhaps (0,0.1], (0.1,0.3], (0.3,0.5], (0.5,0.7], (0.7,0.9], (0.9,1.0]. This keeps the 'end' bins separate as these are more interesting. At present this Fig 4 and also 9 has so much detail and numbers, that the main message is a bit hidden. Finally, it may be worth noting how much data is in these bins. Although the bins are the same size (0.2 in ice fraction for example), the distributions mean there could be relatively few points in some bins.

For wind class bin (9,10], the wind speed greater than 10 m/s is not included. We agree that it is helpful to explain that "(" means exclusive and "]" means inclusive. We added this to the caption of Figure 4:

"In the labels of different sea-ice concentration and 10-m wind speed classes, "(" means

exclusive and "]" means inclusive."

We agree that the boxplot figures in Figure 4 are complex, now we followed the suggestion of the Referee that change the wind class bin size to 2 m/s and rearranged the sea-ice fraction classes to (0,0.1], (0.1,0.3], (0.3,0.5], (0.5,0.7], (0.7,0.9], (0.9,1.0]. Also, we stress that we additionally provide Figures 5, 7, 10, where we display the median values only for an easier visualization of the relationships. The modified Figure 4 is as follow:

[Figure]

**Figure 4**: Box-whisker plots of the relationship between sea-ice drift speed and sea-ice concentration for different near-surface wind speed classes (different colors) for 2003-2014, in the model for (a) winter (DJFM) and (b) summer (JJAS), and in observation/reanalysis data for (c) winter and (d) summer. For the model, all 10 ensemble members are included. The plot is based on daily data and on all grid points within the study domain indicated in Figure 1. The horizontal bar represents the median, the notch represents the 95% confidence interval of the median, the dot represents the mean, the top and bottom of the box represent the 75th and 25th percentiles, the upper/lower whiskers

represent the maximum/minimum value within 1.5 times interquartile range (IQR) to 75/25 percentiles. The numbers above the boxplots represent the slopes of near-surface wind and sea-ice drift speed fit lines (unit: km d$^{-1}$ per 1 m s$^{-1}$ wind speed change; font colors as for the wind speed classes). The numbers right of the boxplots represent the slopes of sea-ice concentration and sea-ice drift speed fit lines (unit: km d$^{-1}$ per 10% sea-ice concentration change). A bold and asterisked number indicates that the slope of the fit line is significant at the 95 % level. In the labels of different sea-ice concentration and 10-m wind speed classes, "(" means exclusive and "]" means inclusive. The sample size of each boxplot is shown in Table 1.

We understand the Referee's concern about the sample size in each bin. Instead of giving the sample size of each bin directly in Figure 4, we provide therein the 95% confidence range of the median value for each bin (represented by the height of the notch in the boxplot). The confidence range includes the influence of the sample size. We provide the sample size for each bin in the new Table 1 as follow:

**Table 1** The sample sizes of sea-ice drift speed data under different 10-m wind speed and sea-ice concentration classes in Figure 4.

| Season | Data source | Wind classes | Sea-ice concentration classes | | | | |
|---|---|---|---|---|---|---|---|
| | | | (0.1,0.3] | [0.3,0.5] | (0.5,0.7] | (0.7,0.9] | (0.9,1.0] |
| DJFM | model | (0,2] m/s | 494 | 864 | 3328 | 135516 | 45562216 |
| | | (2,4] m/s | 2070 | 3716 | 12780 | 489893 | 183056161 |
| | | (4,6] m/s | 3432 | 5766 | 17172 | 618383 | 239287363 |
| | | (6,8] m/s | 5804 | 9402 | 17215 | 435238 | 181532612 |
| | | (8,10] m/s | 7950 | 12511 | 18413 | 246921 | 102066710 |
| | KIMURA/ERA-I/NSIDC | (0,2] m/s | 0 | 7 | 7 | 40 | 102295 |
| | | (2,4] m/s | 15 | 29 | 50 | 124 | 365803 |
| | | (4,6] m/s | 32 | 66 | 117 | 279 | 499634 |
| | | (6,8] m/s | 28 | 88 | 137 | 381 | 386667 |
| | | (8,10] m/s | 53 | 83 | 145 | 288 | 218638 |
| JJAS | model | (0,2] m/s | 92547 | 2535519 | 17432992 | 25896542 | 9142130 |
| | | (2,4] m/s | 322269 | 8521163 | 57295068 | 84426782 | 30714941 |
| | | (4,6] m/s | 534300 | 12186383 | 81681898 | 113048227 | 40015962 |
| | | (6,8] m/s | 549254 | 9864436 | 67161887 | 84920918 | 27346273 |
| | | (8,10] m/s | 356102 | 4746320 | 34131702 | 38211228 | 11159363 |
| | KIMURA/ERA-I/NSIDC | (0,2] m/s | 1519 | 3439 | 5150 | 12096 | 100317 |
| | | (2,4] m/s | 4727 | 10504 | 16204 | 39911 | 312814 |
| | | (4,6] m/s | 6533 | 14571 | 22364 | 55418 | 385667 |
| | | (6,8] m/s | 5261 | 11976 | 17587 | 40559 | 262251 |
| | | (8,10] m/s | 2514 | 5249 | 7276 | 17272 | 107988 |

**Figure 5**

Same comments as above really and note some of the colours are very faint (7,8) class. These plots are more readable but need to be consistent with Fig 4.

Agree and now we enhanced the visualization of Figure 5 by increasing the font, re-arranged into 3×2 panels, reduced the sea-ice concentration and wind speed classes as discussed before and discarded the faint color that previously used for wind class (7,8] m/s. The new Figure 5 is as follow:

[Figure]

**Figure 5**: Relationship between sea-ice drift speed and sea-ice concentration for different near-surface wind speed classes (different colors) in the model during 2003-2014 (a) winter (DJFM) and (d) summer (JJAS). (b) and (e) are

based on observation/reanalysis data. (c) and (f) are based on PIOMAS data. The points in the plot are the median value of all the daily data and on all grid points for certain wind speed and sea-ice concentration, within the study domain indicated in Figure 1.

**Missing Reference:**

Elvidge, A.D., I.A. Renfrew, A.I. Weiss, I.M. Brooks, T.A. Lachlan-Cope, and J.C. King 2016:Observations of surface momentum exchange over the marginal-ice-zone and recommendations for its parameterization, *Atmospheric Chemistry and Physics,* **16,** 1545-1563. doi:10.5194/acp-16-1545-2016

We added the missing reference.

---

## Author Comment (AC2) · 10 Dec 2019

**Answer to the comments of Referee #2**

We would like to thank Referee #2 for his/her suggestions for improving our paper. All the comments have been addressed and point by point response is provided below each comment. The reviewer comments are written in black, our answer in blue, and the corrections in the paper are highlighted in red. The line numbers which are used in the answers correspond to the new version of the manuscript (PDF file) unless otherwise indicated.

**General description**

The Manuscript validates a fully coupled ocean, sea ice and atmospheric circulation model (HIRHAM-NAOSIM). Main focus is on the correlation between the sea-ice drift, sea-ice conditions and the near surface wind speed. The model is validated towards remotely sensed data and another model (PIOMAS). The latter is more a comparison than a validation.

Comparisons has been made at two time scales. First the seasonal variation has been compared, then the daily variations and correlations between mainly ice drift and wind speed in different sea ice regimes.

Results are in general at the level of other model systems, however a few pointers are provided to places where HIRHAM-NAOSIM performs better than PIOMAS Introduction of sea-ice form drag influences the drift speed but this does not improve the overall performance.

**Mayor revisions/concerns**

A paper that validates a model is of relevance, however it seems like there are many references to the 2019 paper by Dorn et al. Without having read this I am a little puzzled whether this manuscript is more of the same of if it points to new findings. Especially when tuning of the form drag is postponed to a later paper.

The paper by Dorn et al. (2019) is cited as reference to the model description and to the long-term simulation setup (BASE run). The study in this paper evaluates sea-ice drift and its dependency on wind speed and sea-ice conditions, which was not addressed by Dorn et al. (2019). Therefore, this study is not a follow-up paper to Dorn et al. (2019), but presents exclusively new findings.

In addition, section 4 presents a sensitivity test for a new parameterization that added the sea-ice form drag from sea-ice edges. The focus is to investigate the sensitivity of simulated sea-ice drift to the new parameterization. We also added a new sub-section 4.2 and new figures to find whether the new parameterization improved the model simulations or not. Fine-tuning of the new parameterization is postponed, since it requires a lot of model simulations. Nevertheless, we added a new sub-section 4.3 that describes the ideas of which parameters could be tuned in the follow-up study to improve the model simulation.

My main concern with this manuscript is that it presents many numbers and correlations

but there is a lack of introduction, perspective and discussion. A few lines is mentioned in the end of section 4 where ocean forcing is mentioned. I think that this should be the start of a discussion that discuss the reasons why for instance the seasonal cycle is poorly represented. How well is the internal ice pressure described? Are observations always the truth? For instance what are the uncertainties/biases of the KIMURA dataset.

We agree that there should be more discussion on our results. We followed the Referee's suggestions and added the uncertainty of KIMURA dataset at line 202:

"The uncertainty of the KIMURA data over the Arctic is from 1.12 to 1.47 km d$^{-1}$ in summer and depends on the drift speed (Sumata et al., 2015a). In winter, the uncertainty is at least 50% smaller than in summer and depends on the drift speed too (Sumata et al., 2015b)"

We also included the discussion about the KIMURA sea-ice drift speed uncertainty and compared it with the model bias at line 276.

"Compared with the uncertainty in the KIMURA sea-ice drift speed provided by Sumata et al. (2015a, b), the model bias in summer is close to or slightly smaller than the uncertainty of the KIMURA data. This indicates that the sign of the model bias in summer SID is uncertain. In winter, however, the model clearly overestimates the SID over the central Arctic and north of the Canada Archipelago and Greenland, even if considering the uncertainty of the KIMURA data."

We added a new paragraph to discuss the reason for wintertime SID overestimation at line 300:

"The overestimation of winter SID could be related to the underestimation of winter SIT (Figure S2). Besides, the prescribed values of the ice-ocean drag coefficient and the ice strength parameter could also play a role. The ice-ocean drag coefficient in the base configuration of HIRHAM-NAOSIM ($5.5 \times 10^{-3}$) is comparable to other CMIP5 models, even though a few models use 3-4 times higher coefficients (see Tandon et al., 2018). Higher ice-ocean drag might damp the SID and its strong dependency on the wind speed. Docquier et al., (2017) show that the higher the ice strength parameter, the lower the winter SID and the higher the winter SIT. The ice strength parameter in the base configuration of HIRHAM-NAOSIM (30,000 N m$^{-2}$) is already slightly higher than in all CMIP5 models (see Tandon et al., 2018). This value has been established for stand-alone ocean-ice simulations with daily wind forcing, but might still be too low considering the hourly wind forcing from the interactively coupled atmosphere."

References:

Sumata, H., Kwok, R., Gerdes, R., Kauker, F., and Karcher, M.: Uncertainty of Arctic summer ice drift assessed by high-resolution SAR data, Journal of Geophysical Research: Oceans, 120, 5285-5301, 2015a.

Sumata, H., Gerdes, R., Kauker, F., and Karcher, M.: Empirical error functions for monthly mean Arctic sea-ice drift, Journal of Geophysical Research: Oceans, 120, 7450-7475, 10.1002/2015jc011151, 2015b.

The article points to the lack of a seasonal cycle and a bad timing of the minimum. My opinion would be that the minimum is more a matter of lack of a seasonal cycle and that this is a random minimum that is irrelevant as long as the seasonal cycle is not present.

A seasonal cycle in the simulated SID is present, even if its amplitude is much weaker than in the KIMURA data due to the model's overestimation of winter SID. We agree that one may argue that it is irrelevant to discuss the minimum as long as the model systematically overestimates the SID just in the season where the observed minimum occurs. On the other hand, the simulated (bad) timing of the minimum is certainly not a random feature, since it appears in all ensemble members and in many other coupled climate models as well. Therefore, we think that it is relevant to point to this model deficit when discussing the seasonal cycle.

From line 52 and the next few lines a method for validation is mentioned. I would recommend to move this into section 2 and describe what this validation method do.

We agree that the previous introduction of the validation method was insufficient and should be located in section 2. We added the following description about the validation method to Section 2.3 at line 233:

"Following Olason & Notz (2014) and Docquier et al. (2017), we use scatter plots showing Arctic basin wide and multi-year averaged monthly mean sea-ice drift speed against sea-ice conditions (sea-ice concentration and thickness) to evaluate the relationships between sea-ice drift speed and sea-ice conditions. The linear fit-lines are added in the scatter plots to assist the comparison of the relationship in the model simulations and in the observation/reanalysis."

Further, we modified the sentence at line 52:

"We first evaluate the simulated Arctic basin-wide monthly mean drift, then we evaluate the relationship between sea-ice drift speed and sea-ice conditions/wind speed both on Arctic basin-wide, multi-year monthly mean scale and on daily grid scale."

Some of the findings are close related. Higher ice drift will lead to lower ice thickness and again higher ice drift. Therefore a comparison with for instance PIOMAS tells you more about the current state of the model than a direct bias (at least that would be my opinion). The comparisons are valid but I will be hesitant to say that for instance the internal strength of the model is too weak. A relevant discussion related to PIOMAS would be to discuss the difference between a forced ocean-sea ice model and a fully coupled model ocean-sea ice-atm model. Are there features that could be described by this?

We agree that the comparison with PIOMAS sea-ice thickness only tells us something about the current state of the model. In order to clarify this, we added one sentence in section 2.2.3 at line 222:

"Therefore, the comparison with PIOMAS informs us rather about the current state of

the model than about a direct model bias."

We also agree that a discussion of the difference between PIOMAS, a forced ocean-sea ice model, and HIRHAM-NAOSIM, a fully coupled atmosphere-ocean-sea ice model, and the possible contribution of this difference to the sea-ice thickness differences between PIOMAS and HIRHAM-NAOSIM should be provided. Therefore, we added the following discussion at line 281:

"The SIT differences between HIRHAM-NAOSIM and PIOMAS could be partly caused by the differences between a fully coupled atmosphere-ocean-sea ice model and a forced ocean-sea ice model. Only the former includes the feedback of the atmosphere to the sea-ice and ocean component."

In addition to PIOMAS, we have considered to use Cryosat2 product for sea-ice thickness, but Cryosat2 is only available from 2010 onwards and does not cover the whole period 2003-2014. Nevertheless, we decided to add the comparison of sea-ice thickness from Cryosat2 and from the model simulations during winter 2010-2014 in supplementary Figure S2. It shows that the sea-ice thickness difference between Cryosat2 and the model is qualitatively similar to the difference between PIOMAS and the model. Therefore, we added the following discussion at line 284:

"Analysis of the SIT differences between HIRHAM-NAOSIM and CryoSat2 during winter 2010-2014 (Figure S2) confirms that HIRHAM-NAOSIM underestimates the SIT over the central Arctic and north of the Canada Archipelago and Greenland, at least in winter."

**Minor corrections**

Line 44: In my opinion the comparison of CMIP 3 models is outdated. The reference provided afterwards is more relevant (Tandon et al 2018).

We agree that the CMIP3 results from Rampal et al. (2011) are not up to date anymore. Therefore, we removed the corresponding statement in Section 1.

Line 50: Please don't start the section with Thus. For instance change to: This paper/manuscript has two aims.

We follow your suggestion, and deleted "Thus" at the beginning of that paragraph.

Line 54. Stating that an observation is rare seems a bit short and subjective. They do exist (RGPS buoys, SAR drift), however these are not present for the entire period. Choosing not to use them is valid but again a few more lines on why would be nice.

We agree and removed the phrase: "Since both model evaluations and observational studies based on the daily grid scale are rare"

Line 75: Replace with: The organization of this paper is as follows: Section 2.

We changed the text as suggested.

Line 84 to 95: A map of the domain and the where the boundaries extend to would improve the understanding of the model domain.

We understand the concern of the Referee about the model domain. However, as the model domain and the study area of interest (shown by the purple line) are already shown in Figure 1, we think there is no need to provide an additional figure. The domain of the ocean-sea ice component is exactly the domain shown in Figures 1, 3, 8, and 11. In the revised version, we added corresponding information to the respective figure captions. Further, we added the following sentence at the end of the model description section 2.1.1 (line 111):

"The ocean-sea ice domain corresponds to the domain shown in Figure 1."

Line 92 reference a dynamic-thermodynamic model described by Harder is an upgrade? What is upgraded. Dynamics are referenced to 1979 and thermodynamics to 1976. Maybe "update" should be removed or explicitly explained what is the update.

We agree that the expression 'upgrade' is confusing. Therefore, we deleted the phrase "and represents an upgrade of the original Hibler (1979) model" in line 118 of the discussion paper and keep the sea-ice model reference of Harder et al. (1998). Compared to Hibler (1979), the sea-ice dynamics include an upstream advection scheme (to avoid negative ice thicknesses), no explicit diffusion, and drag coefficients optimized by comparison with observed buoy drift (Harder and Lemke, 1994; Fischer, 1995; Drinkwater et al., 1995; Harder, 1996; Kreyscher et al., 1997). As these improvements were already mentioned by Harder et al. (1998), we abstained from repeating it again in this paper.

**References**:

Drinkwater, M. R., Fischer, H., Kreyscher, M., & Harder, M. (1995, July). Comparison of seasonal sea-ice model results with satellite microwave data in the Weddell Sea. In 1995 International Geoscience and Remote Sensing Symposium, IGARSS'95. Quantitative Remote Sensing for Science and Applications (Vol. 1, pp. 357-359). IEEE.

Fischer, H. (1995). Vergleichende Untersuchungen eines optimierten dynamisch-thermodynamischen Meereismodells mit Beobachtungen im Weddellmeer= Comparison of an optimized dynamic-thermodynamic sea ice model with observations in the Weddell Sea. Berichte zur Polarforschung (Reports on Polar Research), 166.

Harder, M. (1994). Erweiterung eines dynamisch-thermodynamischen Meereismodells zur Erfassung deformierten Eises. Berichte aus dem Fachbereich Physik, Report, 50.

Harder, M. (1996). Dynamik, Rauhigkeit und Alter des Meereises in der Arktis-numerische Untersuchungen mit einem großskaligen Modell= Dynamics, roughness, and age of Arctic sea ice-numerical investigations with a large-scale model (Doctoral dissertation, Universität Bremen).

Kreyscher, M., Harder, M., & Lemke, P. (1997). First results of the Sea-Ice Model

Intercomparison Project (SIMIP). Annals of Glaciology, 25, 8-11.

Line 104: How is the spinup designed? Running one year 22 times? Has the model bin spun up properly or is the ensemble a representation of the spinup? A bit more elaboration of the choices would be nice. Is Levitus data near the area of interest good enough? Does this imply that the variations seen only originates from the atmosphere?

The BASE ensemble simulations were already carried out for the study by Dorn et al. (2019). The design of the preceding spin-up simulation was described in detail by Dorn et al. (2019): "initial ocean and sea-ice fields were taken from the Januaries 1991 to 2000 of a preceding coupled spin-up run for the period 1979–2000. The coupled spin-up run already reached a quasi-stationary seasonal-cyclic state of equilibrium for the mid-1980s. Consequently, all ensemble members were initialized with ocean and sea-ice fields that represent the diversity of ocean–ice conditions within the steady state of the specific model configuration".

To better emphasize that the BASE ensemble simulations, including the preceding spin-up simulation, were carried out for the study by Dorn et al. (2019), we reformulated the beginning of the paragraph at line 125:

"A 10-member ensemble of multi-decadal climate simulations for the period 1979–2016 were carried out by Dorn et al. (2019) with the base configuration of HIRHAM-NAOSIM 2.0. These multi-decadal ensemble simulations represent the basis for the present study and are referred to as BASE hereafter. The individual BASE ensemble members used the same atmospheric initialization, but applied different ice-ocean initial conditions, which were taken from January 1 of the last 10 years of a preceding 22-year-long coupled spin-up run (see Dorn et al., 2019, for more details)."

We also modified the description of the initial condition for CTRL and SENS simulations at line 192:

"The ice-ocean initial conditions for CTRL and SENS were produced in exactly the same way as for BASE."

The Levitus climatology is only used at the open boundary in the northern North Atlantic (at approx. 50°N). Since this boundary is far away from the area of interest (our study domain in this paper), the Levitus data are not an issue for the present study. Even though there are no externally forced variations at the lateral ocean boundary, there are variations between the ensemble members as well as year-to-year variations. Variations between the ensemble members are by definition a result of internally generated variability in the model. This comprises both atmosphere and ice-ocean variability. In contrast, year-to-year variations in the ensemble mean can be attributed to the external forcing at the lateral atmospheric boundaries (and the surface boundary conditions outside the coupling domain).

Line 157: Validation towards AMSRE. Is the ice drift It would be interesting to see how the model performed vs RGPS buoys and Sentinel 1 SAR icedrift data. Alternatively an evaluation of the uncertainty of the chosen drift product versus the

bias/uncertainty of the model results.

It is beyond of the scope of our study to evaluate KIMURA drift data against buoy and SAR drift data. Intercomparison studies of Arctic ice drift exist in literature. For example, Sumata et al. (2014) intercompared four remotely sensed ice-drift products (incl. KIMURA) and compared them also with available buoy data. Also, there is a whole international activity to validate sea-ice drift products (http://esa-cci.nersc.no/, http://esa-cci.nersc.no/?q=webfm_send/195). It is also beyond the scope of our study to evaluate the model with other data sets such as buoys and SAR data. We have justified in section 2.2.1 why we have selected the KIMURA data set. It is because it has a much wider spatial and temporal coverage than buoys data and is therefore appropriate for regional model evaluation (Sumata et al., 2015a). Another advantage of the KIMURA product is that it provides ice drift data both in winter and summer. More details are given by Kimura et al. (2013) and Sumata et al. (2015a).

Sumata, H., Lavergne, T., Girard-Ardhuin, F., Kimura, N., Tschudi, M. A., Kauker, F., Karcher, M., and Gerdes, R. ( 2014), An intercomparison of Arctic ice drift products to deduce uncertainty estimates, J. Geophys. Res. Oceans, 119, 4887– 4921, doi:10.1002/2013JC009724

According to the Referee's suggestion, we include information about the uncertainty of the KIMURA product and link that with the identified model bias.

We included at line 202:

"The uncertainty of the KIMURA data over the Arctic is from 1.12 to 1.47 km d-1 in summer and depends on the drift speed (Sumata et al., 2015a). In winter, the uncertainty is at least 50% smaller than in summer and depends on the drift speed too (Sumata et al., 2015b)."

And, we included at line 276:

"Compared with the uncertainty in KIMURA sea-ice drift speed provided by Sumata et al. (2015a, b), the model bias in summer is close to or slightly smaller than the uncertainty of the KIMURA data. This indicates that the sign of the model bias in summer SID is uncertain. In winter, however, the model clearly overestimates the SID over the central Arctic and north of the Canada Archipelago and Greenland, even if considering the uncertainty of the KIMURA data."

Line 162: As partly mentioned the comparison with PIOMAS just shows whether NAOSIM provides the same as PIOMAS. Why not use Icesat as mentioned in the discussion about PIOMAS. Admitted there are relatively high uncertaintes on ice thickness products like IceSat, however reference a model and motivate this choice by its skill vs another product seems weird. Other data sets that can be used are operation ice bridge and Cryosat. They do not cover the full period and domain but they can do as Ground Truth.

We followed the Referee's suggestions and added the comparison of sea-ice thickness from the model and from Cryosat2 during 2010-2014 by extending supplementary

Figure 2 and Section 3.1 (line 284):

"Analysis of the SIT differences between HIRHAM-NAOSIM and CryoSat2 during 2010-2014 winter (Figure S2) confirms that HIRHAM-NAOSIM underestimates the SIT over the central Arctic and north of the Canada Archipelago and Greenland, at least in winter."

The modified supplementary Figure 2 is as follow:

[Figure]

**Figure S2:** Mean spatial pattern of sea-ice thickness [m] in the model (ensemble mean) during 2003-2014 (a) winter (DJFM) and (d) summer (JJAS). (b) and (e) are the model differences to the PIOMAS ("Model - PIOMAS") during winter and summer respectively. (c) are the model differences to the CryoSat2 sea-ice thickness in winter during 2010-2014. The purple line in each panel indicates the study domain used for the basin-wide analysis.

Line 187 - 189. Is there a reason for excluding spring and fall?.

The reason for not showing the spring and fall is that we focused our study on the extreme seasons to emphasize the contrast between warm and cold conditions. Figures for spring and fall show intermediate results and do not provide additional insights.

Line 211-216 Not sure why it is required to include such a long description of why sea ice drift is influenced by thickness, concentration and wind speed. This is stated in several articles. Just state that the drift is governed mainly by ice conditions, wind speed and ocean currents (less important).

We agree that the previous description was unnecessarily long. We shorted the sentences at line 266 as follows:

"SID is mainly governed by near-surface wind, sea-ice conditions, and ocean currents."

Line 240 - Small variation of wind don't explain variation of ice drift. The modelled ice drift seems to be controlled mostly by the wind, however this is in contrast to obs.

We agree that the previous wording was misleading. We reworded the paragraph (lines 312) as follows:

"As shown in previous studies (Docquier et al., 2017; Kushner et al., 2018; Olason & Notz, 2014; Tandon et al., 2018), the observed distinct mean seasonal cycle of SID (maximum in autumn, minimum in spring) is obviously not solely controlled by the wind speed, which is strongest in winter and weakest in summer (Figure 2). The phase lag between the seasonal cycle of SID and WS is about 3-4 months in observations/reanalysis (KIMURA ice drift/ERA-I wind). The modeled seasonal cycle and magnitude of the WS agrees well with the ERA-I reanalysis. According to the delayed SID minimum (section 3.1), the phase lag between the simulated seasonal cycle of SID and WS is reduced to about one month, like in many CMIP3 and CMIP5 models (Rampal et al., 2011; Tandon et al., 2018), leading to a higher correlation between SID and WS. This indicates that the modeled SID is much stronger controlled by the wind speed than the observed SID."

245 - 250 Again too high correlated wind and ice drift speed in winter. Other factors/forcing of the dynamics of the sea-ice must impact. A discussion of these would be relevant in a discussion section.

We added a discussion of potential causes for the model bias to Section 3.1 and refer at the end of this paragraph to this discussion:

"As mentioned in Section 3.1, too high sensitivity of the SID to the wind in winter may be related to the underestimated SIT and model parameters governing the sea-ice dynamics."

Line 260. I thought that there is a dynamical forcing between ocean and sea ice everywhere. This should be more specific.

From a model perspective, ocean and sea ice are coupled throughout the model domain. Here we refer to a specific physical coupling process between ocean and sea ice. We agree that more information is helpful and modified the sentences at line 336:

"This could be the result of a dynamical coupling between sea ice and the coastal ocean as suggested by Nakayama et al. (2012): In a coastal ocean covered with sea ice, wind-forced sea-ice drift excites coastal trapped waves and generates fluctuating ocean currents. These ocean currents can enhance the sea-ice drift when the current direction is the same as the wind-driven drift direction."

Line 300 what is the method? Short description please. Same reference is made in

introduction

A description of the method was added to Section 2.3 at line 248 (see above). In this paragraph, we removed the citations and the introduction of the method. The remaining text at line 378 reads:

"Figure 6a shows the relationship between SID and SIC in terms of the mean seasonal cycle."

Line 350 Abrupt end to line.

Thank you very much! The corrected sentence reads now:

"As pointed out by Olason & Notz (2014), the inverse correlation between drift speed and thickness in winter, when the ice concentration is high, is physically plausible, but the inverse correlation in summer, when the ice concentration is lower, is probably only of statistical nature."

Figure 4 and 5 are hard to read. Please increase font size

We agree that Figures 4 and 5 are hard to read. We enhanced the visibility of the two figures by rearrange the panels and increased the font size as suggested. Besides, we changed the wind class bin size to 2 m/s and the sea-ice fraction classes to (0,0.1], (0.1,0.3], (0.3,0.5], (0.5,0.7], (0.7,0.9], (0.9,1.0]. The modified Figure 4 and 5 are as follows:

[Figure]

**Figure 4**: Box-whisker plots of the relationship between sea-ice drift speed and sea-ice concentration for different near-surface wind speed classes (different colors) for 2003-2014, in the model for (a) winter (DJFM) and (b) summer (JJAS), and in observation/reanalysis data for (c) winter and (d) summer. For the model, all 10 ensemble members are included. The plot is based on daily data and on all grid points within the study domain indicated in Figure 1. The horizontal bar represents the median, the notch represents the 95% confidence interval of the median, the dot represents the mean, the top and bottom of the box represent the 75th and 25th percentiles, the upper/lower whiskers represent the maximum/minimum value within 1.5 times interquartile range (IQR) to 75/25 percentiles. The numbers above the boxplots represent the slopes of near-surface wind and sea-ice drift speed fit lines (unit: km d$^{-1}$ per 1 m s$^{-1}$ wind speed change; font colors as for the wind speed classes). The numbers right of the boxplots represent the slopes of sea-ice concentration and sea-ice drift speed fit lines (unit: km d$^{-1}$ per 10% sea-ice concentration change). A bold and asterisked number indicates that the slope of the fit line is significant at the 95 % level. In the labels of different sea-ice concentration and 10-m wind speed classes, "(" means exclusive and "]" means inclusive. The sample size of each boxplot is shown in Table 1.

[Figure]

**Figure 5**: Relationship between sea-ice drift speed and sea-ice concentration for different near-surface wind speed classes (different colors) in the model during 2003-2014 (a) winter (DJFM) and (d) summer (JJAS). (b) and (e) are based on observation/reanalysis data. (c) and (f) are based on PIOMAS data. The points in the plot are the median value of all the daily data and on all grid points for certain wind speed and sea-ice concentration, within the study domain indicated in Figure 1.

---

## Referee Report (RR1)

Review: Evaluation of Arctic sea-ice drift and its dependency on near-surface wind and sea-ice conditions in a coupled regional climate model HIRHAM-NAOSIM

The manuscript describes the evaluation of sea ice drift as function of wind speed, ice concentration and ice thickness in a coupled model. At last a sensitivity study of the implementation of form drag around floe edges is examined. The model system reproduces the summer/autumn drift well but the seasonal variation is missing, thus the winter drift is over estimated.

The manuscript is in general well written and with a few minor changes I think that it is ready for publication

Line 39: I read the line starting wih "The seasonal Arctic..." as the is mainly correlated with ice concentration, thickness and  form drag. These are of course important but the wind speed should also play a role unless the variation is canceled out on longer time period. The statement could be a bit more clear or a reference could be added.

Line 190 The mean horizontal resolution in the Arctic is …. (docquier et al 2017). The refeernce points to a nemo-lim article. Does this include references to Piomas as well? Otherwize why reference this?

Line 197: Providing an uncertainty for the passive microwave product is a bit simplified. The uncertainty/bias depends on proximity to land(not for the evaluation of this model), meltponds and regridding from satellite swats to daily gridded products. In general larger uncertainty near the ice edge and in summer. I would just mention that there are uncertainties due to these factors.

Line 200 as Line 190

Line 203. Even if PIOMAS was not assimilated I would not call the difference a bias of NAOSIM. It is just another model.

Line 282. I assume that this is pstar (30.000 Nm^-2). Some sort of reference or description of the valued

Line 327 ca. I think that it is more correct to use approximately

Lin3 347 Figure 5. Is this figure 5c and 5f?

Lin3 357 and figure 6: Does it make sense to make a fit of all months when the concentration is the same in group 2.. Instead could group 2 have a different colorcoding/symbolsof the dots?

Line 411. See line 327

Line 434 A bit unclear what normalized (none normalized). What are they normalized with?

Section 4.3 I would probably move 4.3 to the end of the summary/conclusion section

Figure 5. I thnk that the y-axis to the left can be removed )model, KIMURA/ERA-Interim,PIOMAS)

Figure 7: References to sub-figure
 d needs to be c in the first and last occurance on line 731